# A renal clearable fluorogenic probe for in vivo β-galactosidase activity detection during aging and senolysis

Sara Rojas-Vázquez [1,2,3,9], Beatriz Lozano-Torres[1,2,4,9],
Alba García-Fernández[1,2,4], Irene Galiana[1,2,4,5], Ana Perez-Villalba [6,7],
Pablo Martí-Rodrigo [3,7], M. José Palop[3,7], Marcia Domínguez[1,2], Mar Orzáez [4,8],
Félix Sancenón[1,2,4,5], Juan F. Blandez[1,2,4,5], Isabel Fariñas [3,7] &
Ramón Martínez-Máñez [1,2,4,5]

Accumulation of senescent cells with age leads to tissue dysfunction and related diseases. Their detection in vivo still constitutes a challenge in aging research. We describe the generation of a fluorogenic probe (sulfonic-Cy7Gal) based on a galactose derivative, to serve as substrate for β-galactosidase, conjugated to a Cy7 fluorophore modified with sulfonic groups to enhance its ability to diffuse. When administered to male or female mice, β-galactosidase cleaves the O-glycosidic bond, releasing the fluorophore that is ultimately excreted by the kidneys and can be measured in urine. The intensity of the recovered fluorophore reliably reflects an experimentally controlled load of cellular senescence and correlates with age-associated anxiety during aging and senolytic treatment. Interestingly, our findings with the probe indicate that the effects of senolysis are temporary if the treatment is discontinued. Our strategy may serve as a basis for developing fluorogenic platforms designed for easy longitudinal monitoring of enzymatic activities in biofluids.

The design and development of new cost-effective and easily implementable diagnostic tools is an important goal in health[1]. In this context, diagnostic systems capable of detecting target biomarkers in readily accessible biofluids constitute a potential solution for non-invasive longitudinal studies[2]. An approach that fulfills these characteristics is the design of probes that can be specifically transformed in cells and tissues and have a rapid renal clearance thus allowing their detection in the urine. A few reports have described the use of multiplexed protease-sensitive nanoparticles that, in response to

proteolytic cleavage in disease environments, release small reporter probes that can be measured in urine by mass spectrometry or immunoassays[3,4]. This approach has been elegantly exploited to detect acute kidney injury with fluorescent and chemiluminescent derivatives equipped with a (2-hydroxypropyl)-β-cyclodextrin (HPβCD) moiety that promoted the renal clearance of the probes[5,6]. Another example describes a nanosensor based on ultra-small renally removable nanoparticles able to recognize deregulated proteases in cells. In this case, a colorimetric signal is indirectly detected by measuring the ability of

[1]Instituto Interuniversitario de Investigación de Reconocimiento Molecular y Desarrollo Tecnológico (IDM), Universitat Politècnica de València-Universitat de València, Valencia, Spain. [2]CIBER de Bioingeniería, Biomateriales y Nanomedicina (CIBER-BBN), Valencia, Spain. [3]Instituto de Biotecnología y Biomedicina (BIOTECMED), Universitat de València, Valencia, Spain. [4]Unidad Mixta UPV-CIPF de Investigación en Mecanismos de Enfermedades y Nanomedicina, Universitat Politècnica de València, Centro de Investigación Príncipe Felipe, Valencia, Spain. [5]Unidad Mixta de Investigación en Nanomedicina y Sensores, Universitat Politècnica de València, IIS La Fe, Valencia, Spain. [6]Laboratory of Animal Behavior Phenotype (L.A.B.P.). Facultad de Psicología, Universidad Católica de Valencia, Valencia, Spain. [7]Centro de Investigación Biomédica en Red de Enfermedades Neurodegenerativas (CIBERNED), Valencia, Spain. [8]Centro de Investigación Príncipe Felipe, Valencia, Spain. [9]These authors contributed equally: Sara Rojas-Vázquez, Beatriz Lozano-Torres. ✉e-mail: isabel.farinas@uv.es; rmaez@qim.upv.es

the urine-recovered nanoparticles to oxidize a chromogenic perox-idase substrate in the presence of hydrogen peroxide[7]. These systems, however, rely on the application of complex and expensive analytical assays, or use nanoparticles which could result in undesired accumu-lation or side effects[8]. A reasonable alternative could be based on the use of a dye or fluorophore in an OFF state whose turning ON is dependent on the presence of a specific biomarker, such as an enzyme, that interacts with the probe at the site of the disease and that is chemically designed to favor its diffusion out of the cells and its fil-tration by the kidney into the urine. However, such a simple idea has not been widely exploited[9,10].

Aging is characterized by a progressive and generalized functional decline and by the concomitant development of age-related diseases[11]. One of the hallmarks of aging is a rise in the frequency of senescent cells in most organs. Cells undergo senescence in response to several stressors, entering an irreversible cell cycle arrest that is accompanied by an increase in heterochromatin and in foci of DNA damage-responsive proteins, or DNA scars, increased levels of cyclin-dependent kinase (CDK) inhibitor proteins p16$^{INK4a}$ (p16) and/or p21$^{CIP1}$ (p21), and reduced levels of nuclear lamin B1. They also exhibit larger cell size and an augmented lysosomal compartment, with highly increased levels of the enzyme β-galactosidase (β-Gal)[12]. Senescent cells also display an intense secretory activity which appears to be involved in the pathophysiology of many aging related diseases through paracrine effects on neighbor healthy cells[13,14]. Senolysis, or the selective pharmacological killing of senescent cells, is thus being explored as a potentially promising intervention to promote tissue rejuvenation and health[15,16]. Within this context, there is a pressing need for strategies that focus on the in vivo identification of senescent cell load and its continual tracking over time, both in the aging process and following senolytic interventions[17-20]. Current non-invasive pro-cedures rely primarily on the use of near-infrared fluorescence and MRI probes to detect increased β-Gal activity by in vivo imaging[21-24].

Here, we report the design and synthesis of a cyanine-7-based probe and its ability to act as an in vivo read-out of β-Gal activity in biofluids without the necessity of imaging techniques. This probe consists of a Cy7 fluorophore modified with two $SO_3$ groups and conjugated to a β-Gal substrate. Upon in vivo administration, the non-emissive sulfonic-Cy7Gal probe becomes hydrolyzed by β-Gal in cells to release the highly fluorescent dye sulfonic-Cy7[25]. Subsequently, the fluorophore diffuses out of the cells and is quickly cleared at the kid-neys, allowing its detection and quantitation in the urine. We show that the probe can be used to detect cell senescence in mice non-invasively and that it provides readouts during aging and senolytic intervention that correlate with age-associated anxious behavior. Interestingly, our in vivo studies reveal the transient nature of senolytic treatments. The swift optical determination of overall β-Gal activity through a simple urine measurement represents an attractive alternative to other pro-cedures and enables easy longitudinal studies in aging research. Our results also indicate that the same strategy could be used to generate fluorogenic probes to monitor the activity of disease-related enzymes in easily collectable biological fluids in experimental animals and, eventually, in humans.

## Results

### Design and synthesis of a probe to monitor in vivo β-Gal activity in biofluids

Fluorescent probes that are turned ON from an OFF state by the hydrolysis of an intramolecular O- or N-glycosidic bond by β-Gal at lysosomes have been previously generated by our group and others for detecting senescent cells[19,21,24,26]. Building on this work, we deci-ded to develop a sulfonic-probe based on a Cy7 fluorophore skeleton modified with two $SO_3$ groups and conjugated to a galactose deri-vative, with the idea that the addition of sulfonic groups to the releasable fluorophore would promote the diffusion out of the cells

and its recovery in the urine when injected in vivo[27]. To pursue this goal, we first decided to synthesize the probe without this sulfonic modification to assess its value as a specific detector for β-Gal activity inside senescent cells. The probe without sulfonic groups (WOS-Cy7Gal) was prepared following a two-step synthetic procedure shown in Fig. 1a. First, 2,3,4,6-tetra-O-acetyl-α-D-galactopyranosyl bromide was reacted with 4-hydroxyisophthalaldehyde in anhydrous acetonitrile yielding compound **1**. Then, a Knoevenagel condensation between **1** and 1-butyl-2,3,3-trimethyl-3H-indol-1-ium iodide (**2**) yiel-ded the WOS-Cy7Gal probe. Moreover, the WOS-Cy7 fluorophore was synthesized by protecting the hydroxyl group of 4-hydroxyisophthalaldehyde with t-butyldimethylsilyl chloride fol-lowed by a Knoevenagel condensation with **2** and the subsequent deprotection of the hydroxyl group. WOS-Cy7Gal and WOS-Cy7 were fully characterized by $^1H$-NMR, $^{13}C$-NMR, and HRMS (see Methods). PBS (pH 7) solutions of the WOS-Cy7 fluorophore showed an intense emission at ca. 660 nm ($\Phi_{WOS-Cy7} = 0.48$) when excited at 580 nm, whereas PBS solutions of the WOS-Cy7Gal probe were poorly emis-sive at the same excitation wavelength ($\Phi_{WOS-Cy7Gal} = 0.0076$).

In order to validate the ability of WOS-Cy7Gal to monitor cell senescence, we tested the probe in cultures of primary venous endo-thelial cells obtained from the human umbilical cord (hUVECs) treated with 1 μM palbociclib, a CDK4/6 inhibitor that reportedly induces cell cycle arrest and senescence[28]. Treatment with the drug for 7 days resulted in enlargement of the cells, which exhibit a more flattened morphology and nuclear alterations, in agreement with morphological features described for senescent phenotypes[29]. The induction of cell senescence was corroborated by increased β-Gal activity in the X-Gal histochemical reaction (Fig. 1b). Treated cells ceased proliferation (Ki67-negative) and exhibited increased levels of DNA damage (γH2AX-positive foci) and p21 protein and lower levels of lamin B1, all cell senescence features[12] (Fig. 1c, Supplementary Fig. 1a). Pharmacologically-induced senescent hUVECs were strongly positive for WOS-Cy7-emission after 10 min in the presence of the probe (Fig. 1d). Likewise, we could detect the emission of the probe when these cells reached replicative senescence after 29 passages (Supple-mentary Fig. 1b, c). Our results indicated that WOS-Cy7Gal is an appropriate probe to monitor senescence in cells.

Subsequently, we set out to correlate the probe detection with well-established markers of cell senescence at the single cell level. Given that the β-Gal enzyme is still active after fixation, we formaldehyde-fixed palbociclib-treated and untreated hUVECs before the immunostaining for p16 and lamin B1. Next, we applied WOS-Cy7Gal and monitored its emission signal after only 10 min. Increased levels of emission from punctate structures resembling lysosomes in the senescent cells were found to co-label with apparently lower lamin B1 levels and higher p16 levels (Fig. 1e, f). To establish a quantitative correlation between p16 and probe-related fluorescent levels, we immunostained fixed cells in suspension with antibodies to p16 and then exposed the cells to WOS-Cy7Gal for 10 min, and both fluorescent emissions were monitored 15 min afterwards by flow cytometry. We observed that the cells with higher levels of p16 were also brighter for the WOS-Cy7 emission (Fig. 1g, h), indicating that the WOS-Cy7Gal probe can be a reliable indicator of senescence-associated β-Gal activity.

We next synthesized the same probe with sulfonic groups added. A Knoevenagel condensation between **1** and, in this case, 1-(4-sulfo-butyl)−2,3,3-trimethylindolium inner salt (**3**) yielded the sulfonic-Cy7Gal probe (Fig. 2a). The sulfonic-Cy7 fluorophore was synthe-sized in the same manner as in the case of WOS-Cy7. The probe and the fluorophore were fully characterized by $^1H$-NMR, $^{13}C$-NMR, and HRMS (see Methods section). PBS (pH 7) solutions of the sulfonic-Cy7 fluor-ophore showed an intense broad emission band centered at ca. 660 nm ($\Phi_{Cy7} = 0.43$) when excited at 580 nm, whereas PBS solutions of the sulfonic-Cy7Gal probe were poorly fluorescent at the same

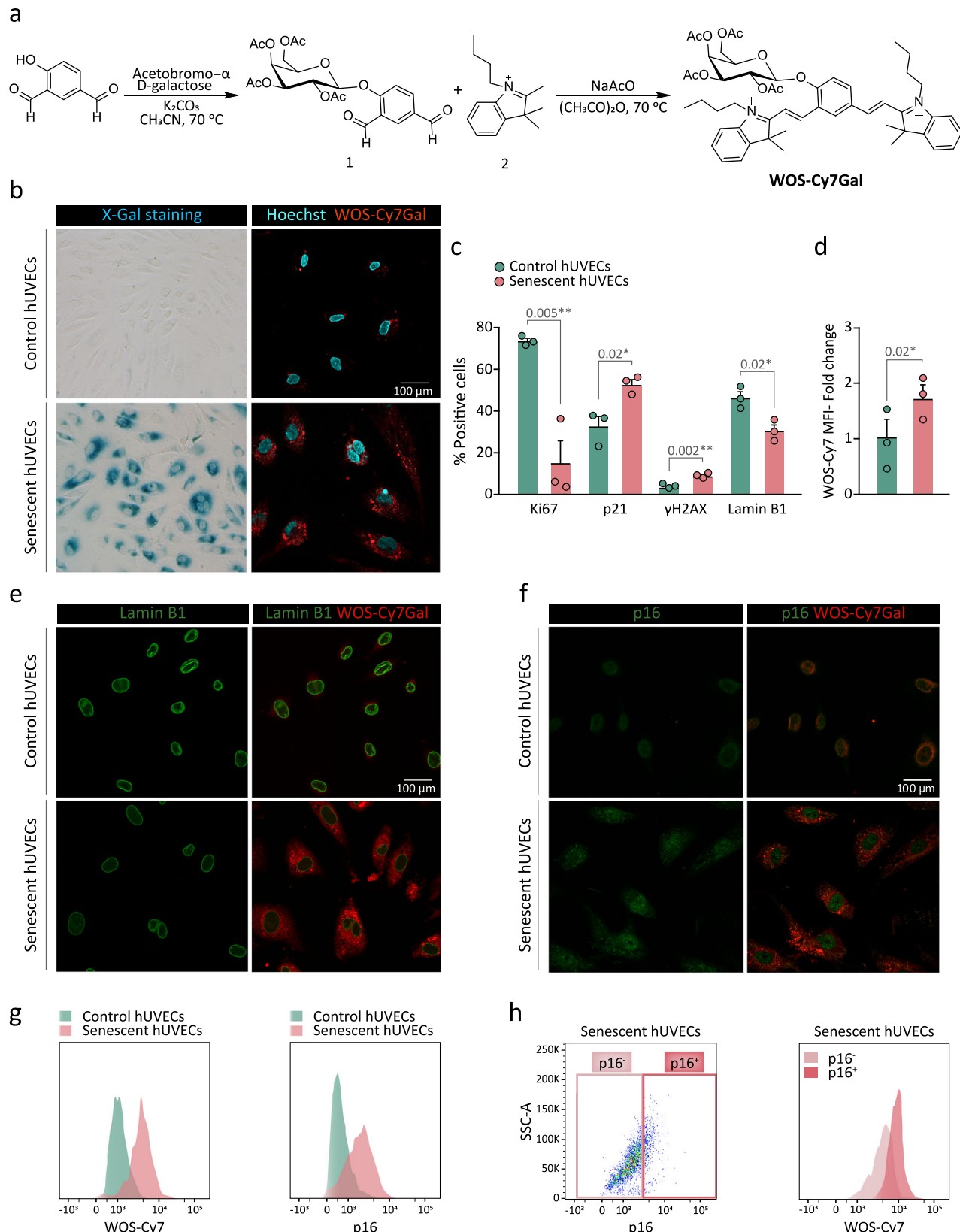

excitation wavelength ($\Phi_{Cy7Gal}$ = 0.0062) (Supplementary Fig. 2a). The fluorescence emission intensity of sulfonic-Cy7 fluorophore remained unchanged in the 5-10 pH range (Supplementary Fig. 2b, c). The hydrolysis of sulfonic-Cy7Gal in PBS solutions in the presence of the β-Gal enzyme was studied by HPLC. The chromatograms showed the disappearance of the sulfonic-Cy7Gal peak and the appearance of

sulfonic-Cy7 if the enzyme was present (Supplementary Fig. 2d). The specificity and selectivity of the probe for β-Gal were further tested by incubation of sulfonic-Cy7Gal with different enzymes and with interfering species, such as cations, anions, and small peptides (Supplementary Fig. 2e). Only β-Gal induced a marked emission enhancement at ca. 660 nm due to the hydrolysis of sulfonic-Cy7Gal. Interestingly, an

**Fig. 1 | Design, synthesis and validation of WOS-Cy7Gal for SA-β-Gal activity detection. a** Synthetic route used for the preparation of the WOS-Cy7Gal probe. **b** X-Gal histochemical staining (left) and confocal images of WOS-Cy7Gal (right) in hUVECs, treated with the senescence-inducing drug palbociclib (senescent) or not (control), for the determination of SA-β-Gal activity. **c** Phenotypic characterization of senescence in hUVECs by immunocytochemical detection of senescence-associated markers (n = 3 control and senescent cells). Note that senescent hUVECs showed a marked decrease in proliferation with lower Ki67 levels, a significant increase in the levels of p21 and DNA damage foci γH2AX, as well as a loss of lamin B1. **d** Quantification of WOS-Cy7Gal-associated median fluorescence intensity (MFI) in control and senescent hUVECs by flow cytometry (n = 3 control and senescent cells). The fold change refers to control hUVECs. **e** Confocal images of control and senescent hUVECs labeled with lamin B1 and WOS-Cy7Gal. Notice that senescent

cells displayed lower fluorescence levels of lamin B1 and higher of the Cy7 fluorophore, compared to control ones. **f** Confocal images of control and senescent hUVECs labeled with p16 and WOS-Cy7Gal. Observe that senescent cells showed a brighter signal for both markers compared to control ones. **g** Flow cytometry histogram of the MFI of probe-released Cy7 and of p16, comparing control and senescent hUVECs. **h** Percentage of p16$^{-/low}$ and p16$^{+/high}$ senescent cells and flow cytometry histogram of WOS-Cy7Gal-associated MFI within these two cell populations. It is worth noting that the highest levels of fluorescence related to WOS-Cy7Gal corresponds to p16$^{+/high}$ cells. The graphs show the mean ± SEM. Unpaired and paired Student's two-tailed t-tests were used for statistical analysis in graphs c and d, respectively. The number of independent biological samples (represented as dots) used and the exact p-values are indicated in the graphs. Scale bars: 100 μm. Source data is provided as a Source Data file.

enhanced emission was observed when the sulfonic-Cy7Gal was incubated in the presence of both esterase and β-Gal, but not esterase alone (Supplementary Fig. 2e). The signal increase is likely due to the hydrolysis of the acetate moieties in the sulfonic-Cy7Gal probe by esterases prior to the rupture of the O-glycosidic bond by β-Gal, a sequence that is expected to happen intracellularly. Both sulfonic-Cy7Gal and WOS-Cy7Gal yielded brighter emission in naturally senescent *vs*. replication-competent hUVECs in culture but, as expected, the signal with sulfonic-Cy7Gal exhibited a more diffuse cellular pattern that was compatible with lower retention in lysosomes (Supplementary Fig. 2f, g).

To evaluate whether sulfonic-Cy7Gal could indeed provide a reliable readout in urine of cell senescence in vivo, we used it in mice in which the senescent cell burden could be experimentally manipulated. To this end, we decided to produce grafts of a controlled number of senescent cells using the 4T1 mouse mammary tumor cell line, a cellular model of triple negative-breast cancer that is sensitive to palbociclib-induced senescence[30-32]. We first corroborated that 4T1 cells enter senescence in vitro when treated with 5 μM palbociclib for 2 weeks as reported. The treatment resulted in increased β-Gal activity in the X-Gal histochemical reaction and higher levels of β-Gal protein in immunoblots (Supplementary Fig. 3a). Subsequently, control and senescent 4T1 cells were incubated with 20 μM sulfonic-Cy7Gal and analyzed by confocal microscopy 2 h post-incubation (Supplementary Fig. 3a). Neither control nor palbociclib-treated 4T1 cells exhibited any noticeable fluorescence signal in the absence of the probe when excited at 580 nm. After exposure to sulfonic-Cy7Gal, control 4T1 cells still displayed negligible fluorescence emission, whereas senescent 4T1 cells showed a 2.4-fold higher red emission that could be significantly reduced (ca. 60%) if cells were pre-incubated for 30 min with the β-Gal specific inhibitor D-galactose at 5 mM (Supplementary Fig. 3b). A marked reduction in the intensity was also found when the expression of *Glb1*, the gene that encodes lysosomal β-Gal, was interfered with a specific *Glb1* siRNA (Supplementary Fig. 3c, d), indicating that β-Gal is responsible for the emission. Finally, viability assays indicated that the probe was innocuous for both normal and senescent cells (Supplementary Fig. 3e).

We next implanted orthotopic grafts of 1×10⁶ 4T1 cells into the left mammary fat pad of BALB/cByJ young female mice that were subsequently treated by daily oral gavage with 0, 10, 50, or 100 mg/kg palbociclib for 7 days to induce different degrees of cell senescence in the grafts. Tumors from mice treated with 10 mg/kg of palbociclib grew similarly to those in untreated mice, while tumors in mice treated with 50 or 100 mg/kg palbociclib displayed a significant reduction in volume, measured with a caliper (Supplementary Fig. 4a). Histological analyses indicated a progressively reduced proportion of Ki67-positive cells as the palbociclib dose increased, in line with its reported induction of cell senescence[33,34], and increased β-Gal activity in the tumors, but not in organs such as liver or kidney (Supplementary Fig. 4b–d), providing us with a model of controlled cell senescence load in vivo. Next, mice bearing 4T1 tumors and treated with

palbociclib at the different concentrations were anesthetized and intraperitoneally (i.p.) injected with 2.5 μmol of sulfonic-Cy7Gal or WOS-Cy7Gal. In vivo IVIS® imaging after 15 min revealed a detectable fluorescent signal in the bladders of mice injected with sulfonic-Cy7Gal, which increased with the palbociclib dosage (Fig. 2b and Supplementary Fig. 5a, b, for ROI selection and quantification in the bladder). The same amount of the WOS-Cy7Gal probe did not generate any fluorescent signal in the bladder of mice bearing 4T1 tumors and treated with 100 mg/kg palbociclib (Fig. 2b and Supplementary Fig. 5a, b, for ROI selection and quantification in the bladder). Urine subsequently recovered from the same animals was fluorescent only in those injected with sulfonic-Cy7Gal and also in a palbociclib dose-dependent manner (Fig. 2c and Supplementary Fig. 5c, d). Fluorometer measurements, relative to a sulfonic-Cy7 calibration curve, revealed a clear correlation of the sulfonic-Cy7 μmol levels un urine with the palbociclib dosage (Fig. 2d). In agreement with a swift renal clearance of the sulfonic-Cy7 fluorophore in animals injected with the sulfonic-Cy7Gal probe, we found a detectable amount of sulfonic-Cy7 in the urine (ca. 2.34 μmol) by fluorometer measurements, while the presence in plasma was significantly lower (0.33 μmol). In contrast, the WOS-Cy7 fluorophore was basically undetectable either in plasma (0.02 μmol) or urine (0.01 μmol) in animals injected with the WOS-Cy7Gal probe (Fig. 2e). A simple mass balance of the amount of sulfonic-Cy7Gal injected and that of sulfonic-Cy7 in urine measured allowed us to calculate that, on average, ca 95% of injected sulfonic-Cy7Gal was excreted through urine as sulfonic-Cy7 in palbociclib treated mice (100 mg/kg) while the excretion of WOS-Cy7 from WOS-Cy7Gal was negligible (ca 1%).

To corroborate the results at the organ level, we analyzed ex vivo the tumors, the kidneys and the bladder of those mice treated with 100 mg/kg palbociclib after euthanasia. In these autopsy samples, both probes gave a positive signal in the tumors (Fig. 2f, g), in accordance with their capacity to detect senescent cells. In contrast, we could detect a strong fluorescent signal in the bladder only in the animals injected with the sulfonic-Cy7Gal probe (Fig. 2f, g). The lack of an equivalently strong signal in the kidneys suggests a rapid clearance of the fluorophore, in line with previously shown results in plasma (Fig. 2e). Additional postmortem analyses in the tumors and different organs of the animals treated with increasing palbociclib doses and injected with the sulfonic-Cy7Gal indicate that the progressive rise in the fluorescence in urine derives from the experimentally produced senescence load in the tumors (Supplementary Fig. 6a, b). Our results together demonstrate that sulfonic acid moieties promote a rapid renal clearance of the Cy7 fluorophore upon cleavage of the sulfonic-Cy7Gal probe by the β-Gal enzyme.

## The sulfonic-Cy7Gal probe is a reliable urine detector for the aging-associated increase in β-Gal activity

We next aimed to test the potential of the sulfonic-Cy7Gal probe for monitoring overall β-Gal activity during aging by i.p. injecting it in healthy 2- and 14-month-old (m) BALB/cJyB mice. IVIS® images of the

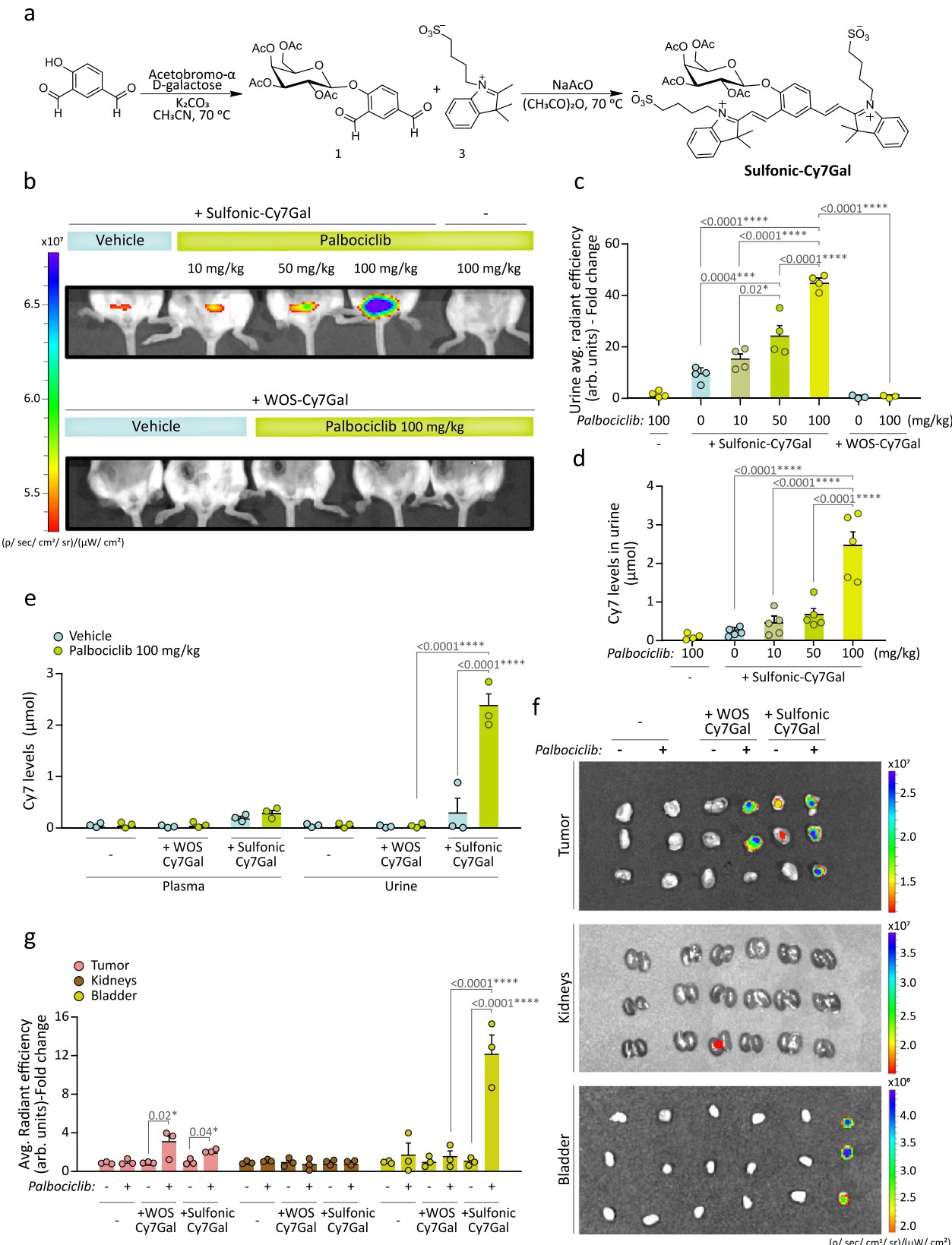

anesthetized mice 15 min post-injection revealed fluorescence accumulation in the bladder of 14-m, but not 2-m mice (Fig. 3a and Supplementary Fig. 7a, b for ROI selection and quantification in the bladder). The fluorescent signal detected by IVIS® imaging in the urine collected from elderly mice was 5.2-fold stronger than from young mice (Fig. 3b), in agreement with an expected higher systemic β-Gal

activity in old animals[13]. After euthanasia, the bladder, brain and lungs were studied also by IVIS® imaging. Quantification of the emission intensity revealed a 5.2-fold increase in the bladder of aged *vs*. young mice and increases of 2.3 and 2.7-fold for brain and lungs, respectively, in aged mice when compared to young animals (Supplementary Fig. 7c–e). The latter is in agreement with increases in cell senescence

**Fig. 2 | In vivo SA-β-Gal selectivity and renal clearance of the sulfonic-Cy7Gal probe. a** Synthetic route used for the preparation of the sulfonic-Cy7Gal probe, carrying sulfonic groups. **b** At the top, in vivo IVIS® imaging of sulfonic-Cy7Gal-associated fluorescence in bladders of BALB/cByJ female mice bearing 4T1 mammary tumors and orally treated with palbociclib (0, 10, 50, 100 mg/kg), compared to mice treated with the highest dose of palbociclib but non-injected with the probe. At the bottom, the same animal model injected with the WOS-Cy7Gal probe, comparing those receiving 100 mg/kg of palbociclib to untreated mice. **c** IVIS® readout of sulfonic-Cy7 average radiant efficiency in urine from mice under the conditions described in b (n = 4 for sulfonic-Cy7Gal injected-mice, n = 4 for non-probe-injected mice and n = 3 for WOS-Cy7Gal injected-mice). Note that the level of fluorescence increases in urine as a function of palbociclib dose. The fold change refers to untreated sulfonic-Cy7Gal-injected mice. **d** Quantification of the amount (μmol) of sulfonic-Cy7 fluorophore excreted in urine by fluorometer measurements after i.p. injection of sulfonic-Cy7Gal, using a calibration curve

'(n = 5 biologically independent animals). **e** Quantification of the fluorophore recovered after i.p. injection of sulfonic-Cy7Gal or WOS-Cy7Gal in urine and plasma from mice bearing 4T1 tumors and treated or not with palbociclib (100 mg/kg) using a fluorometer and a calibration curve (n = 3 biologically independent animals). See that only the probe carrying sulfonic groups is rapidly recovered in urine. **f** Ex vivo IVIS® imaging of tumors, kidneys and bladders from breast tumor-bearing mice, treated or not with 100 mg/kg of palbociclib, and injected or not with WOS-Cy7Gal and sulfonic-Cy7Gal. **g** Readout of the probes-associated fluorescence with IVIS® in the organs and experimental conditions exposed in image **f** (n = 3 biologically independent animals). The fold change refers to untreated mice for each organ and probe. Fluorescence (average radiant efficiency) associated with each ROI is measured in arbitrary units (arb.units). The graphs show the mean ± SEM. One-way ANOVA and Tukey's multiple comparison post-hoc tests were used for statistical analysis. Source data is provided as a Source Data file.

---

incidence in lungs reported with aging[35]. In the brain, β-Gal activity is not a specific marker of senescence, as many healthy neurons have large lysosomal compartments[36]. Interestingly, senescent-like neurons and glial cells have been identified in mouse models of neurodegeneration and in neuropathological human tissue[37,38]. The apparent capacity of the probe to permeate the blood-brain barrier, however, opens the possibility to study global β-Gal activity, including that of the brain. Furthermore, aging-associated endothelial cell senescence can increase BBB permeability[39] potentially contributing to the use of the probe for studies in brain aging and neurodegeneration. Because renal function can vary depending on the mouse strain[40], we also tested the sulfonic-Cy7Gal probe in mice of the C57BL/6 strain and found again higher fluorescence in the urine from old 15-m animals relative to 3-m younger counterparts by fluorimeter-based detection (Fig. 3c). Direct measurement of the relative levels of the probe and the released fluorophore in the urine by HPLC indicated that most of the sulfonic-Cy7Gal probe is excreted intact in young animals (ca. 80%) whereas the sulfonic-Cy7 fluorophore is the major species excreted in aged mice (ca. 75%) (Fig. 3d). These results were indicative of the potential of the sulfonic-Cy7Gal probe to monitor aging.

We decided to also use senescence accelerated mouse (SAM) strains which exhibit premature aging traits. A few decades ago, intensive inbreeding of AKR/J mice and selection for the early appearance of features such as hair loss, skin coarseness, and short life span, led to the isolation of senescence-prone (P) and senescence-resistant (R) series of mice which were crossed separately to establish the inbred SAMP and SAMR strains[41]. Relative to their genetic background-controls (SAMR1 mice), SAMP8 mice manifest several aging traits at earlier physiological ages[41] and are widely used in aging research to study immune dysfunction[42], osteoporosis[43] or brain atrophy[44]. Injection of the sulfonic-Cy7Gal and WOS-Cy7Gal probes in 15-m mice of the SAMP8 background also resulted in the detection of fluorescence in urine, and in plasma to a lesser extent, only in animals treated with the renally-clearable probe (Fig. 3e).

Because the age-related phenotypic differences between SAMP8 and SAMR1 mice begin to be evident after approximately 6 months of age[45–47], we injected 7-m SAMR1 and SAMP8 mice with sulfonic-Cy7Gal and found a stronger fluorescent signal in the urine of SAMP8 vs. SAMR1 mice (Fig. 3f). We next decided to test whether the sulfonic-Cy7Gal probe could indeed be providing an in vivo readout of cell senescence, by comparing urine fluorescence with markers of senescence at the cell level in SAM mice. After euthanasia, we extracted the kidneys and livers, and their isolated cells were incubated ex vivo with the cell-trapped WOS-Cy7Gal probe or intracellularly immunostained for p16 or lamin B1 prior to their analysis by flow cytometry. In agreement with the levels of fluorescence in the urine of the animals, we found higher levels of β-Gal activity in SAMP8 vs. SAMR1 mice with the WOS-Cy7Gal probe (Fig. 4a). Likewise, we could observe higher proportions of cells with detectable levels of p16 and reduced levels of

lamin B1 in cell dissociates from the organs of SAMP8 mice (Fig. 4b, c). These data indicated a good correlation between the end-point measurement of cell senescence with well-accepted markers and the in vivo global detection of β-Gal activity with the sulfonic-Cy7Gal probe in urine.

## The use of the sulfonic-Cy7Gal probe reveals that senolytic effects are transient

As the urine excretion of the sulfonic-Cy7 fluorophore enables longitudinal studies, we next set out to determine whether the sulfonic-Cy7Gal probe could be used to reliably monitor the effects of a senolytic treatment. Combination of dasatinib (D), an inhibitor of several tyrosine-kinases used as an anti-neoplastic agent for the treatment of acute lymphocytic leukemia, and quercetin (Q), a flavonoid that acts as an anti-apoptotic BCL-XL protein inhibitor, has shown efficacy in counteracting mechanisms of apoptosis evasion in senescent cells resulting in their selective elimination and subsequent beneficial effects in aged organs[16,48]. In our senolytic experiment, 15-m C57BL/6 mice were treated by oral gavage with two doses of D (5 mg/kg) + Q (50 mg/ml) or vehicle (20% PEG-n400 in saline solution) per week for 5 weeks and, 13 days after the end of the treatment, they were injected with the sulfonic-Cy7Gal probe (Fig. 5a). We could observe lower fluorescence levels in the urine of treated vs. untreated mice (Fig. 5b).

The selected senolytic treatment reportedly alleviates brain phenotypes associated with aging and neurodegeneration[49,50]. Accordingly, as another non-invasive measure of the senolytic treatment, we decided to perform parallel behavioral tests that reportedly correlate with age-related declines in mice. Because an increase in anxiety is associated with aging and frailty, we used the open field and the elevated plus maze (EPM) tests to evaluate anxiety-related behavior. The open field test measures overall locomotor activity but also anxiety-like behavior as anxious mice typically avoid the lit central area and spend more time at the periphery of the open box, close to the walls. In the EPM, mice are confronted with the decision to spend time exploring the closed arms or the anxiety-generating open arms of an elevated maze[51,52]. As expected, the time spent in the central area of the open field or in the open arms of the EPM was significantly reduced in 17-m C57BL/6 mice when compared to young 3-m mice of the same strain (Fig. 5c–f). Regarding the senolytic experiment, the animals were tested for their performance in the open field and EPM tests 2 days after the measurement of the probe level in urine (Fig. 5a). The treatment resulted in improved performance of the elderly mice in both behavioral tests (Fig. 5c–f), reflecting lower anxiety after D + Q treatment. A group of 3-m mice was included as behavioral controls without changes in general locomotion (Supplementary Fig. 8). Importantly, we found a significant negative correlation between the sulfonic-Cy7 fluorescence levels in urine and the time spent in the central area of the open field or in the open arms of the EPM (Fig. 5g, h), indicating that animals with an organismal lower β-Gal readout display less anxious

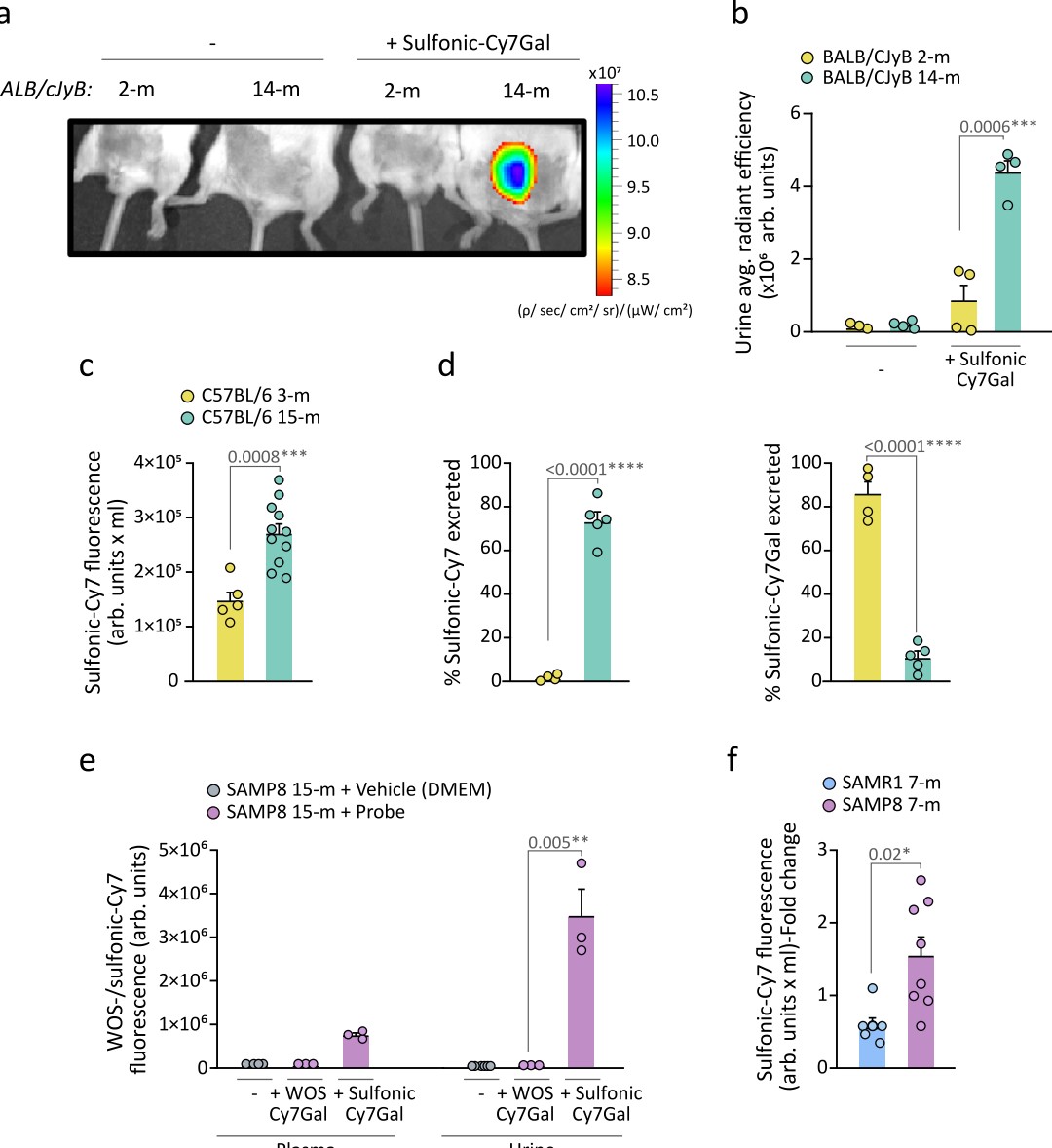

**Fig. 3 | Sulfonic-Cy7Gal-based monitoring of age-related senescence in vivo during natural and accelerated aging. a** In vivo IVIS® imaging of bladders from 2- and 14-m BALB/cByJ mice injected or not with the sulfonic-Cy7Gal probe. **b** IVIS® readout of sulfonic-Cy7 fluorescence in urine from 2- (n = 4) *vs.* 14-m (n = 4) BALB/cByJ mice. **c** Fluorometer measurement of sulfonic-Cy7 associated fluorescence in the urine of 3- (n = 5) and 15-m (n = 11) C57BL/6 mice. Note that this value is multiplied by the total volume of urine recovered (arb.units x mL) to avoid variability in micturition between young and old mice that may alter the concentration of the fluorophore in the bladder and the measurement of fluorescence in urine. **d** Measurement of the percentage of sulfonic-Cy7Gal or sulfonic-Cy7 excreted in

the urine of 3- (n = 4) *vs.* 15-m (n = 5) C57BL/6 mice by HPLC. **e** Fluorescence readout of urine and plasma samples from 15-m SAMP8 mice (n = 3) after i.p. injection of WOS-/sulfonic-Cy7Gal compared to vehicle-injected SAMP8 mice (n = 3) using a fluorometer. **f** Fluorometer measurement of sulfonic-Cy7 fluorescence in the urine of 7-m SAMP8 (n = 8) and SAMR1 mice (n = 6). Fold change is calculated relative to SAMR1 mice values. The graphs show the mean ± SEM. Unpaired Student's two-tailed t-test was used for statistical analysis. The number of independent biological samples (represented as dots) used and the exact p-values are indicated in the graphs. Source data is provided as a Source Data file.

behavior. Interestingly, senescent cell clearance also alleviates obesity-induced anxiety[53]. These data suggested that sulfonic-Cy7Gal is sensitive enough to predict age-related behavior and, more importantly, that this probe can be used to monitor senolytic treatments in vivo.

Next, we decided to test whether our probe could monitor intermittent senolytic treatment. Given that, in comparison with aged SAMR1 mice, the aged SAMP8 mice also show increased anxious behavior in the open field at ages when they still exhibit normal locomotion (i.e., 4 and 9-m) (Supplementary Fig. 9a–c)[41,54], we exposed 7-m SAMP8 mice to the D + Q treatment for 5 weeks and evaluated fluorophore recovery in urine 21 days after cessation of the treatment

(Fig. 6a). As before, we found that the treatment effectively decreased sulfonic-Cy7 levels in the urine (Fig. 6b; first cycle, short-term). A 3-day treatment with the drug combination did not change the behavior of 3-m SAMR1 control mice in the open field test, indicating that the D + Q drug combination per se does not affect anxiety (Supplementary Fig. 9d); however, we observed a decrease in anxious behavior when we assessed the animals at short-term after the senolytic treatment (Supplementary Fig. 9e, f), in line with the reduced sulfonic-Cy7 levels in the urine. We then repeated the measurements at 58 days post-treatment and found no differences in anxiety (Supplementary Fig. 9e) or urine fluorescence between animals treated with D + Q or vehicle (Fig. 6c;

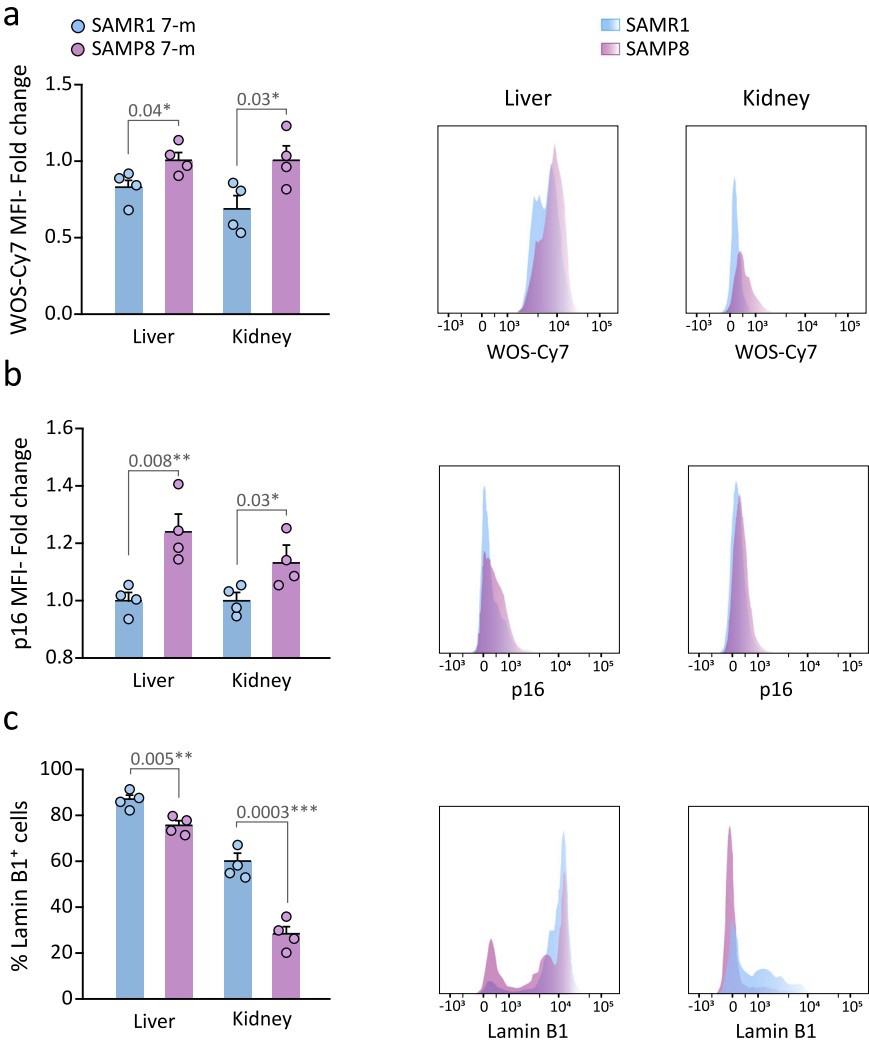

**Fig. 4 | Sulfonic-Cy7 fluorescence readout in the urine of SAM mice correlates with other senescence markers. a** Ex vivo flow cytometry analysis of β-Gal activity based on WOS-Cy7Gal-associated MFI in liver and kidney tissues of 7-m SAMP8 and SAMR1 mice and its histogram representation. **b** Ex vivo flow cytometry analysis of p16 MFI in liver and kidney tissues of SAMP8 and SAMR1 mice and representative flow cytometry histogram. **c** Ex vivo flow cytometry analysis of the percentage of lamin B1+ cells in liver and kidney tissues of SAMP8 and SAMR1 mice and representative histogram of lamin B1 MFI. Fold change is calculated relative to SAMR1 mice organs for each senescence marker. The graphs show the mean ± SEM. Unpaired Student's two-tailed t-test was used for statistical analysis. The number of independent biological samples is n = 4 for SAMP8 and for SAMR1 mice. The exact p-values are indicated in the graphs. Source data is provided as a Source Data file.

first cycle, long-term), suggesting that transient senolysis does not result in permanent alleviating effects. To make sure that the animals could still respond to senolysis, we treated the same animals once more and analyzed them at 16 days after the treatment, finding again the reduction in urine fluorescence (Fig. 6d; second cycle, short-term). As with naturally aged mice of the C57BL/6 strain, we could observe a significant negative correlation between levels of sulfonic-Cy7 fluorescence in urine and the time that SAMP8 mice spent in the central area of the open field test, but only in those short-term cases, i.e. in which the effect was measured less than 3 weeks after senolytic administration (Fig. 6e, f). In terms of sulfonic-Cy7Gal safety in vivo, 7 SAMP8 vehicle mice were injected at 7-m before the senolytic intervention and again at 8, 10 and 11-m for assessing the treatment. On the other hand, 6 C57BL/6 mice were injected with the probe at 15-m, before senolysis, and then at 17-m. All of them survived the entire senolytic longitudinal study, with no unforeseen health issues, until the moment of euthanasia at 12-m in SAMP8 and 18-m in C57BL/6 mice. In the absence of systematic toxicity assays in mice, these observations convey the idea that the probe is not overtly toxic for animals. Together, our data reveals that the probe reliably monitors senolysis and that the effects of

D + Q wash off with time, in line with a treatment that eliminates senescent cells but does not eliminate the inductors of cell senescence.

## Discussion

Geroscience is a very active field in biomedicine. Monitoring the rate of aging requires effective ways to measure systemic changes that can be used as biomarkers of decline progression. An increased prevalence of cell senescence seems to be linked to the deterioration of various organs, yet pinpointing exclusive markers for cell senescence remains a challenging endeavor. Among the available markers, lysosomal β-Gal enzymatic activity stands out as one of the most commonly used indicators of cell senescence[12,55]. Consequently, various probes have been recently devised to provide a means of detecting this state within isolated cells, biopsies, or whole organisms through imaging techniques[17–21]. It is worth noting that senescence-associated β-Gal activity is not due to a specific β-Gal enzyme or a different lysosomal pH, but a higher activity resulting from increased lysosomal mass and increased expression of *Glb1*, the gene encoding the lysosomal β-Gal enzyme. Therefore, the design of a probe specifically tailored to exclusively detect β-Gal activity of senescent cells is difficult to achieve

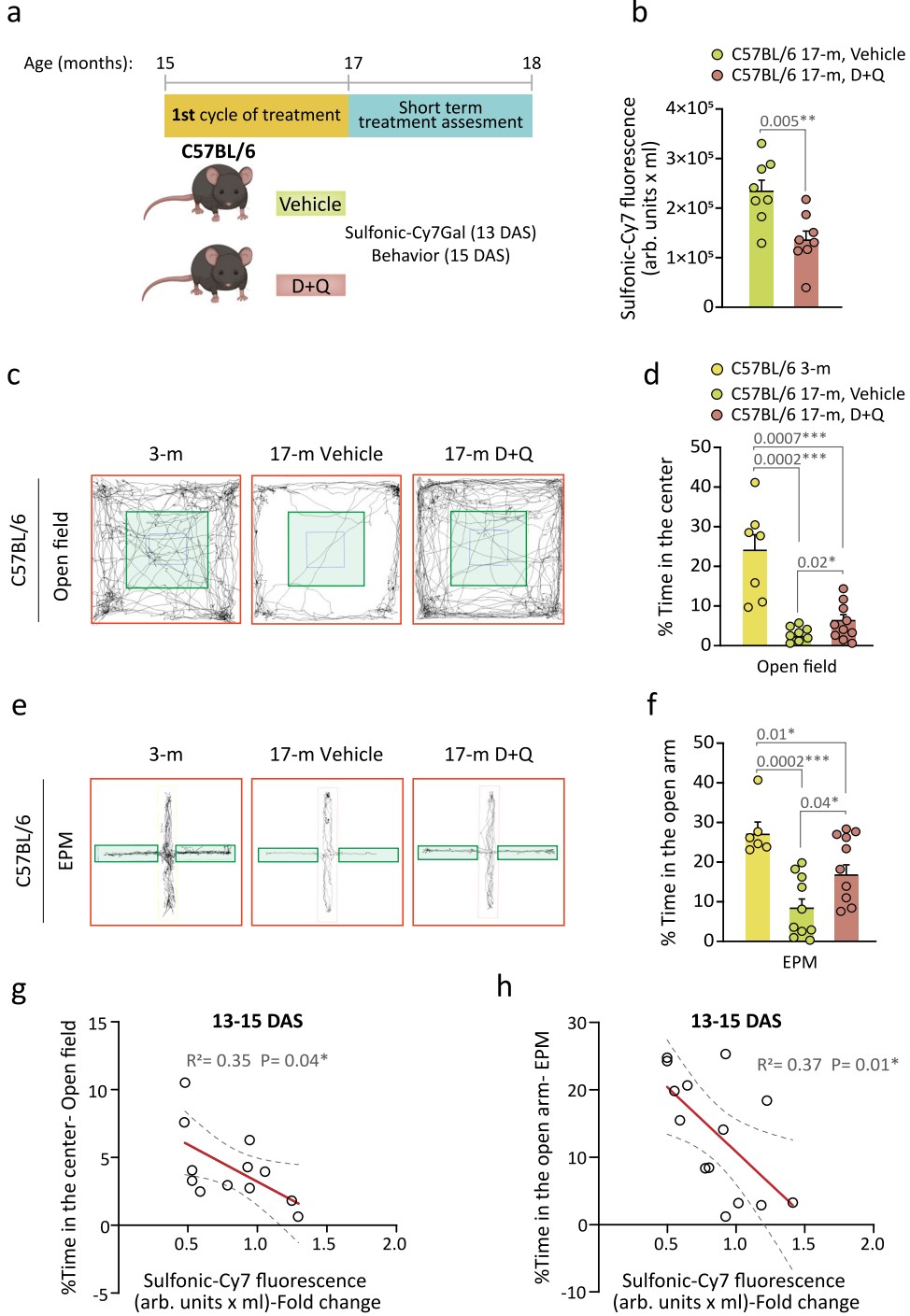

**Fig. 5 | Monitoring of senolytic intervention with the sulfonic-Cy7Gal probe during natural aging. a** Schedule of the senolytic treatment and its monitoring in C57BL/6 mice. 15-m mice received the senolytic drugs D + Q or the vehicle for 5 weeks and, 13 days after senolysis (DAS), the overall β-Gal activity was assessed with sulfonic-Cy7Gal as a measure of senescence burden. Two days later, anxious behavior was examined. **b** Sulfonic-Cy7 fluorescence measurement in the urine of 17-m C57BL/6 treated with D + Q (n = 8) or vehicle (n = 8). **c** Representative map of movement in the open field test comparing young (3-m) to old mice (17-m) that were treated or not with senolytic drugs. **d** Quantification of the percentage of the time that mice spent in the central area (green zone) of the open field, comparing mice aged 3- (n = 7) to 17-m treated with D + Q (n = 10) or vehicle (n = 8). **e**, Representative map of movement in the EPM test comparing young (3-m) to old (17-m) mice that were treated with vehicle or senolytic drugs. **f** Quantification of the percentage of time that mice spent in the open arm of the EPM, comparing mice of

3- (n = 6) and 17-m that were treated (n = 10) or not (n = 10) with D + Q. **g** Significant linear correlation between sulfonic-Cy7 fluorescence in urine and the percentage of time spent in the center of the open field in mice treated with D + Q (n = 5) or vehicle (n = 7). **h** Significant linear correlation between sulfonic-Cy7 fluorescence in urine and the percentage of time spent in the open arm of the EPM test in mice treated with D + Q (n = 6) or vehicle (n = 8). Fold change refers to sulfonic-Cy7 fluorescence levels in the urine of untreated mice in both correlation graphs. Graphs represent mean ± SEM. Unpaired two-tailed Student's t-test was used for statistical analysis of the sulfonic-Cy7Gal fluorescence readout, one-way ANOVA for the assessment of anxious behavior and multiple linear regression to determine the relationship equation between both. P-values and the number of independent biological samples (represented as dots) are shown in the graphs. Source data is provided as a Source Data file.

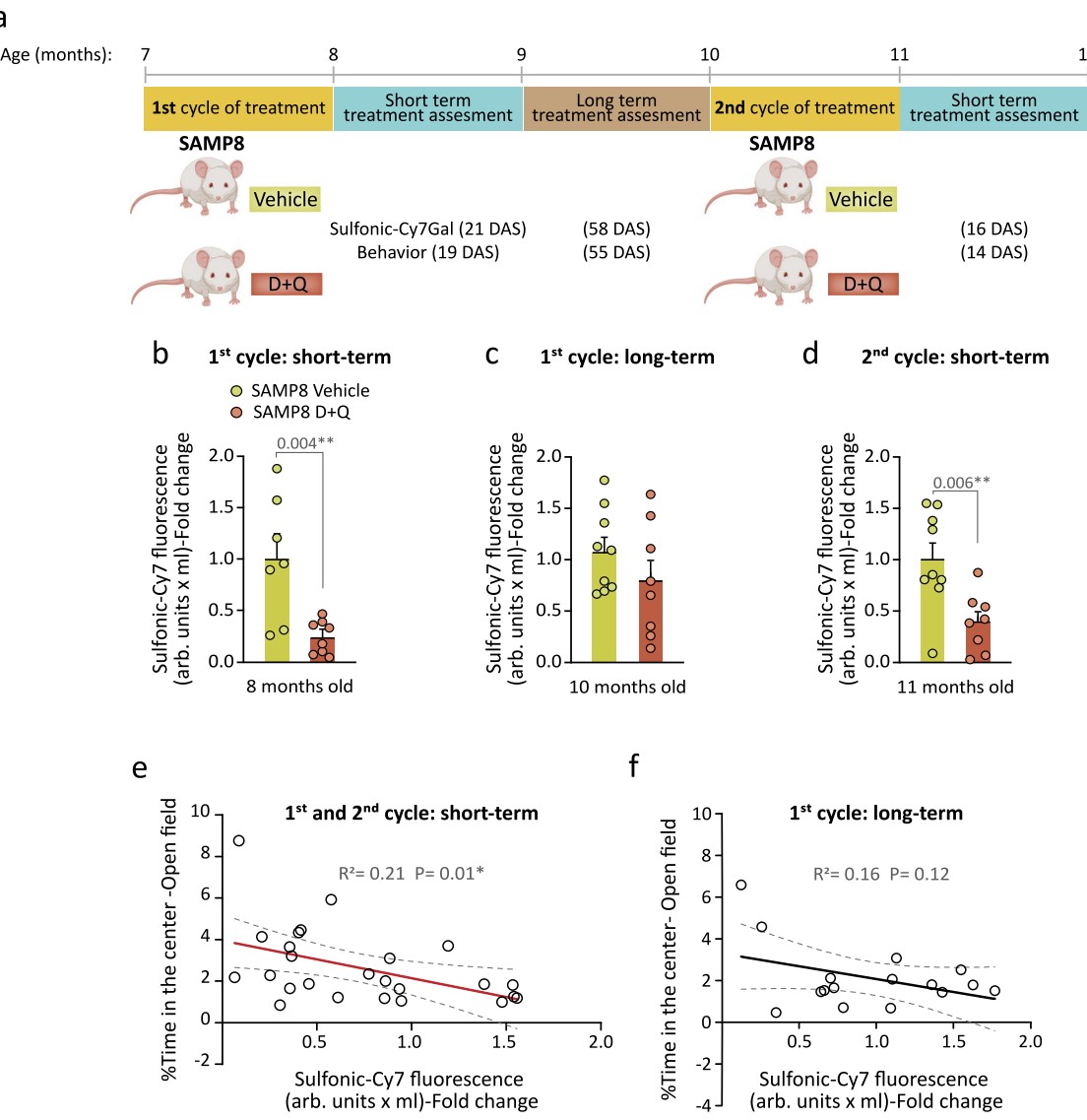

**Fig. 6 | Monitoring of senolytic treatment with the sulfonic-Cy7Gal probe during accelerated aging. a** Schedule of the senolytic treatment and its monitoring in SAMP8 mice. At 7-m, SAMP8 mice received a first cycle of oral treatment with the senolytic drugs D + Q or vehicle, and anxious behavior (open field test) and senescence burden (sulfonic-Cy7Gal) were assessed shortly (first 21 DAS; first cycle, short-term) and long after this treatment (55-58 DAS; first cycle, long-term). In addition, a second cycle was carried out at the age of 10-m and sulfonic-Cy7Gal and behavior were analyzed within the first 16 DAS (second cycle, short-term). **b** Sulfonic-Cy7Gal-associated fluorescence measurement of urine samples from SAMP8 mice, administered with D + Q (n = 8) or vehicle (n = 7), 21 days after the first treatment cycle. **c** Sulfonic-Cy7Gal-associated fluorescence measurement of urine samples from SAMP8 mice, treated (n = 8) or not (n = 9) with D + Q, 58 days after the first treatment cycle. **d** Readout of sulfonic-Cy7Gal-associated fluorescence in urine samples from

SAMP8 mice, treated with D + Q (n = 8) or vehicle (n = 9), 16 days after the second treatment cycle. **e** Significant linear correlation found shortly after treatment between the levels of sulfonic-Cy7 fluorescence in urine and the performance in the open field test of mice treated with D + Q (n = 10) or vehicle (n = 15). **f** No significant correlation was found between sulfonic-Cy7Gal and anxious behavior when assessed in the long term after the senolytic intervention (n = 8 for vehicles and n = 8 for treated mice). Fold change is calculated relative to sulfonic-Cy7 fluorescence in the urine of vehicle mice in all graphs. The graphs represent mean ± SEM. Unpaired two-tailed Student's t-test was used for statistical analysis and multiple linear regression to determine the equation of relationship between sulfonic-Cy7 fluorescence in urine and the time mice spent in the center of the open field. Exact p-values and the number of independent biological samples (represented as dots) are shown in the graphs. Source data is provided as a Source Data file.

and most of the probes currently available rely on the recognition of increased levels of β-Gal enzymatic activity in these cells[56]. A second challenge so far is to develop a biomarker that can be easily monitored. One promising approach in the field involves detecting biomarkers from easily accessible biofluids using the simplest possible detection systems[57,58]. In this context, one innovative strategy pursues the design of fluorogenic probes (in an OFF state) that can be transformed by the action of certain biomarkers (i.e. an enzyme) in cells and tissues to give a final fluorescent product (in an ON highly emissive state) that can diffuse out of the cells and has a rapid renal clearance, thus allowing its detection in the urine using a simple fluorimeter[59]. Based on this

concept, we report herein the case of detection of lysosomal β-Gal using the fluorogenic probe sulfonic-Cy7Gal. We show that cell-entrapped (WOS-Cy7Gal) and diffusible (sulfonic-Cy7Gal) forms of this probe produce a brighter emission in cells undergoing replicative or induced senescence. We provide evidence of the reliability of the sulfonic-Cy7Gal probe to monitor an experimentally controlled load of cellular senescence in vivo by using BALB/cByJ mice bearing breast cancer tumors treated with senescence-inducing chemotherapy. We also show that the fluorophore intensity in urine correlates with age progression and age-associated anxiety in mice during natural and accelerated aging. In addition, the probe allows a non-invasive monitoring of the

effects of senolytic treatment. Interestingly, our data reveals that senolytic treatments based on the pharmacological use of agents that eliminate senescent cells are transient. These findings demonstrate that renal clearable fluorogenic probes are a versatile modular tool that opens new opportunities to develop simple diagnoses in urine for a variety of diseases where abnormal enzymatic activity is a biomarker. We anticipate that this technology can also be applied in the monitoring of therapeutic treatments. Such adaptive detection system could also be applicable in low-resource environments and might democratize access to advanced and sensitive diagnoses.

## Methods

Our research complies with all relevant ethical regulations. All animal procedures were approved by Ethics Committees for Research and Animal Welfare (CEBA) and conducted in accordance with the recommendations of the Federation of European Laboratory Animal Science Associations (FELASA).

### Synthesis and characterization of fluorophores and probes

**WOS-Cy7Gal.** 4-hydroxyisophthalaldehyde (Tokyo Chemical Industry Co. Ltd.; 75 mg, 0.5 mmol), α-acetobromogalactose (Sigma-Aldrich; 607 mg, 1.5 mmol) and potassium carbonate (Sigma-Aldrich; 400 mg, 4 mmol) were mixed in a round bottom flask under argon atmosphere and dissolved in 30 mL of anhydrous acetonitrile (Acros Organics). The reaction mixture was heated to 70 °C and stirred for 4 hours under argon atmosphere and the solvent was removed under vacuum. The residue was purified by column chromatography on silica gel (hexane-ethyl acetate 2:1 v/v as eluent) to obtain product **1** as a yellow solid (230 mg, 0.46 mmol, 92.6% yield). Then, product **1** (90 mg, 0.17 mmol) 1-butyl-2,3,3-trimethyl-3H-indol-1-ium iodide (**2**, 120 mg, 0.35 mmol) and sodium acetate (47.80 mg, 0.35 mmol) were mixed in a Schlenk flask under argon atmosphere and dissolved in 2 mL of acetic anhydride. The reaction mixture was heated to 70 °C and stirred under argon atmosphere. After 4 hours, the solvent was removed under vacuum and the crude product was purified in a silica flash column using diethyl ether as eluent. WOS-Cy7Gal was obtained as red-brown oil (50 mg, 0.05 mmol, 33% yield).

$^{1}$H NMR (400 MHz, CD$_3$OD) δ = 9.15 (d, J = 1.9 Hz, 1H), 8.72 (d, J = 16.1 Hz, 1H), 8.58 (d, 16.2 Hz 1H), 8.44 (dd, J = 8.5; 1.7 Hz, 1H), 8.10 (d, J = 16.1 Hz, 1H), 8.02 (d, J = 16.2 Hz, 1H), 7.85-7.73 (m, 8H), 7.58 (d, J = 8.5 Hz, 1H), 5.75 (d, J = 8.1 Hz, 1H), 5.65 (t, J = 8.7 Hz, 1H), 5.59 (d, J = 3.3 Hz, 1H), 5.35 (dd, J = 9.8; 3.2 Hz, 1H), 4.81-4.73 (m, 4H), 4.35 (t, J = 6.3 Hz, 1H), 4.15 (d, J = 6.2 Hz, 2H), 2.30-2.15 (m, 4H), 2.13 (s, 3H), 2.09 (s, 3H), 2.07-2.01 (m, 4H), 1.98 (s, 3H), 1.94 (s, 3H), 1.89 (s, 3H), 1.85 (s, 6H), 1.83-1.70 (m, 6H), 1.69 (s, 3H).

$^{13}$C- NMR (100 MHz, CD$_3$OD) δ = 185.92 (1 C), 185.87 (1 C), 172.82 (1 C), 171.55 (1 C), 171.41 (1 C), 171.11 (1 C), 153.62 (1 C), 144.64 (1 C), 143.58 (1 C), 143.12 (1 C), 142.95 (1 C), 142.33 (1 C), 139.24 (1 C), 134.42 (1 C), 133.41 (1 C), 132.16 (1 C), 132.11 (1 C), 127.81 (1 C), 126.84 (1 C), 125.36 (1 C), 125.13 (1 C), 124.94 (1 C), 124.67 (1 C), 121.73 (1 C), 115.87 (1 C), 115.55 (2 C), 114.31 (1 C), 113.92 (1 C), 99.45 (1 C), 73.26 (1 C), 70.72 (1 C), 69.43 (1 C), 68.25 (1 C), 61.33 (1 C), 53.35 (1 C), 53.12 (1 C), 52.93 (1 C), 52.78 (1 C), 33.54 (1 C), 33.23 (1 C), 31.76 (1 C), 31.17 (1 C), 26.38 (1 C), 26.11 (1 C), 22.83 (1 C), 22.69 (1 C), 21.65 (1 C), 21.58 (1 C), 21.47 (1 C), 21.35 (1 C), 18.21 (1 C), 18.05 (1 C).

HRMS: calculated for C$_{52}$H$_{64}$N$_2$O$_{10}$ (M + H$^+$) 876.459 m/z; measured 876.7992 m/z (M + H$^+$).

**Sulfonic-Cy7Gal.** Compound **1** (90 mg, 0.17 mmol), 1-(4sulfobutyl)-2,3,3-trimethylindolium inner salt (**3**) (Sigma-Aldrich; 100 mg, 0.35 mmol) and sodium acetate (Sigma-Aldrich; 47.80 mg, 0.35 mmol) were dissolved in 2 ml of acetic anhydride (Sigma-Aldrich) in a Schlenk flask under an argon atmosphere. The reaction mixture was heated to 70 °C and stirred for 4 hours under argon atmosphere and solvent was eliminated under vacuum. The crude was purified by reverse phase column chromatography (dicholoromethane-methanol 10:1 v/v as eluent) to obtain sulfonic-Cy7Gal as a red-brown solid. (120 mg, 0.116 mmol, 68.1% yield).

$^{1}$H NMR (400 MHz, CD$_3$OD) δ = 9.09 (d, J = 1.7 Hz, 1H), 8.60 (d, J = 16.4 Hz, 1H), 8.52 (d, 16.3 Hz 1H), 8.36 (dd, J = 8.9; 1.8 Hz, 1H), 8.04 (d, J = 16.4 Hz, 1H), 7.94 (d, J = 16.3 Hz, 1H), 7.72-7.62 (m, 8H), 7.47 (d, J = 9.0 Hz, 1H), 5.80 (d, J = 7.8 Hz, 1H), 5.58 (t, J = 8.6 Hz, 1H), 5.54 (d, J = 3.5 Hz, 1H), 5.40 (dd, J = 10.4; 3.5 Hz, 1H), 4.92-4.86 (m, 4H), 4.49 (t, J = 6.5 Hz, 1H), 4.20 (d, J = 6.1 Hz, 2H), 3.07-2.98 (m, 4H), 2.34-2.20 (m, 4H), 2.19 (s, 3H), 2.11 (s, 3H), 2.10-2.06 (m, 4H), 2.05 (s, 3H), 2.02 (s, 3H), 1.95 (s, 3H), 1.91 (s, 6H), 1.87 (s, 3H).

$^{13}$C- NMR (100 MHz, CD$_3$OD) δ = 183.88 (1 C), 183.83 (1 C), 171.94 (1 C), 171.80 (1 C), 171.46 (1 C), 171.28 (1 C), 154.75 (1 C), 145.76 (1 C), 145.52 (1 C), 142.26 (1 C), 142.15 (1 C), 142.13 (1 C), 138.32 (1 C), 133.56 (1 C), 132.39 (1 C), 131.36 (1 C), 131.06 (1 C), 128.78 (1 C), 126.18 (1 C), 124.29 (1 C), 124.26 (1 C), 124.14 (1 C), 124.08 (1 C), 122.75 (1 C), 116.65 (1 C), 116.54 (2 C), 116.29 (1 C), 116.22 (1 C), 98.70 (1 C), 72.66 (1 C), 71.92 (1 C), 69.83 (1 C), 68.51 (1 C), 62.43 (1 C), 54.26 (1 C), 54.23 (1 C), 54.12 (1 C), 54.09 (1 C), 52.09 (1 C), 50.86 (1 C), 32.16 (1 C), 32.13 (1 C), 30.71 (1 C), 30.10 (1 C), 27.14 (1 C), 26.72 (1 C), 23.53 (1 C), 23.29 (1 C), 20.79 (1 C), 20.70 (1 C), 20.54 (1 C), 20.50 (1 C).

HRMS: calculated for C$_{52}$H$_{63}$N$_2$O$_{16}$S$_2$ (M + H$^+$) 1035.3619 m/z; measured 1035.3606 m/z (M + H$^+$), 1051.3546 (M + H$_2$O-2H$^+$).

**Sulfonic-Cy7.** 4-hydroxyisophthalaldehyde (1.5 g, 9.9 mmol) and imidazole (1.7 g, 25 mmol) were dissolved in 100 mL of anhydrous dichloromethane. Tertbutyldimethylsilyl chloride (1.8 g, 12 mmol) was then added dropwise to the solution. The reaction was stirred at room temperature (RT) overnight. Afterwards, the mixture was washed with brine (2 ×100 ml) and the organic phase was dried with MgSO$_4$. The solvent was removed under vacuum and the crude was purified by a silica column using a mixture of hexane:ethyl acetate (10:1) as eluent. Hydroxyl-protected product was obtained as colorless oil (2.48 g, 9.39 mmol, 94.8% yield). Subsequently, hydroxyl-protected product (79.24 mg, 0.3 mmol) was added in a Schlenk tube with sodium acetate (83.9 mg, 1.02 mmol) and 1-(4-sulfobutyl)-2,3,3-trimethylindolium inner salt (**3**). The mixture was dissolved in acetic anhydride under argon atmosphere. The reaction was stirred at 80 °C for 4 hours. Acetic anhydride was removed under vacuum and treated with dichloromethame and the solvent was eliminated again under vacuum. Finally, deprotection of hydroxyl group was assessed by stirring the obtained product in a solution of KF·2H$_2$O (47 mg, 0.5 mmol) in acetonitrile (2 ml) for 2 h at RT. Then, the mixture was treated with saturated NaHCO$_3$ water solution (1 ml) and extracted with CH$_2$Cl$_2$ (3 ×20 ml). The organic layer was washed with brine and dried with MgS$_5$O$_4$. The crude was purified by reverse phase column chromatography (dicholoromethane-methanol 8:1 v/v as eluent). Sulfonic-Cy7 was obtained as a brown-green solid (85 mg, 0.120 mmol, 40 % yield).

$^{1}$H NMR (400 MHz, CD$_3$OD) δ = 7.17 (s, 1H), 7.16 (d, J = 8.3 Hz, 1H), 7.11-7.00 (m, 4H), 6.97 (d, J = 10.1 Hz, 1H), 6.82-6.68 (m, 2H), 6.62 (d, J = 9.0 Hz, 1H), 6.59-6.53 (m, 2H), 5.84 (d, J = 10.3 Hz, 1H), 5.31 (d, J = 10.3 Hz, 1H), 4.56 (d, J = 10.2 Hz, 1H), 3.59-3.53 (m, 4H), 2.87-2.81 (m, 4H), 1.87-1.80 (m, 4H), 1.80-1.74 (m, 4H), 1.62 (s, 3H), 1.47 (s, 3H), 1.26 (s, 3H), 1.15 (s, 3H).

$^{13}$C- NMR (100 MHz, CD$_3$OD) δ = 177.40 (1 C), 173.50 (1 C), 173.30 (1 C), 161.40 (1 C), 156.55 (1 C), 154.91 (1 C), 148.99 (1 C), 146.61 (1 C), 144.92 (1 C), 142.64 (1 C), 141.37 (1 C), 139.20 (1 C), 137.86 (1 C), 136.12 (1 C), 122.48 (1 C), 122.42 (1 C), 121.48 (1 C), 119.99 (1 C), 119.65 (1 C), 118.13 (1 C), 115.71 (1 C), 107.54 (1 C), 106.24 (1 C), 106.20 (1 C), 53.07 (1 C), 52.71 (1 C), 52.49 (1 C), 52.41 (1 C), 52.39 (1 C), 45.67 (1 C), 30.00 (1 C), 29.30 (1 C), 28.89 (1 C), 28.84 (1 C), 26.44 (1 C), 26.22 (1 C), 23.80 (1 C), 23.76 (1 C).

HRMS: calculated for C$_{38}$H$_{44}$N$_2$O$_7$S$_2$ (M + H$^+$) 705.269 m/z; measured 705.2673 m/z (M + H$^+$).

**WOS-Cy7**. 4-hydroxyisophthalaldehyde (500 mg, 3.3 mmol) and imidazole (567 mg, 8.3 mmol) were dissolved in 100 mL of anhydrous dichloromethane. Tertbutyldimethylsilyl chloride (603 mg, 4 mmol) was then added dropwise to the solution. The reaction was stirred at RT overnight. Afterwards, the mixture was washed with brine (2 ×100 ml) and the organic phase was dried with MgSO$_4$. The solvent was removed under vacuum and the crude was purified by silica column using a mixture of hexane-ethyl acetate (10:1 v/v) as eluent. The hydroxyl-protected product was obtained as colorless oil (806 mg, 3.05 mmol, 92.5% yield). Then, the hydroxyl-protected product (90 mg, 0.17 mmol), 1-butyl-2,3,3-trimethyl-3H-indol-1-ium iodide (**2**, 120 mg, 0.35 mmol) and sodium acetate (47.80 mg, 0.35 mmol) were mixed in a Schlenk flask under argon atmosphere and dissolved in 2 mL of acetic anhydride. The reaction mixture was heated to 70 °C and stirred under argon atmosphere. After 4 hours, the solvent was removed under vacuum and the crude product was purified in a silica flash column using diethyl ether as eluent. **WOS-Cy7** was obtained as red-brown oil (61 mg, 0.06 mmol, 36 % yield).

$^1$H NMR (400 MHz, CD$_3$OD) δ = 7.15 (s, 1H), 7.13 (d, J = 8.1 Hz, 1H), 7.09-7.01 (m, 4H), 6.92 (d, J = 9.4 Hz, 1H), 6.71-6.59 (m, 2H), 6.55 (d, J = 9.2 Hz, 1H), 6.52-6.41 (m, 2H), 5.81 (d, J = 10.1 Hz, 1H), 5.25 (d, J = 10.2 Hz, 1H), 4.48 (d, J = 10.1 Hz, 1H), 3.50-3.43 (m, 4H), 1.91-1.83 (m, 4H), 1.77-1.69 (m, 4H), 1.68 (s, 3H), 1.49 (s, 3H), 1.40-1.25 (m, 6H), 1.23 (s, 3H), 1.17 (s, 3H).

$^{13}$C- NMR (100 MHz, CD$_3$OD) δ = 176.48 (1 C), 172.56 (1 C), 172.24 (1 C), 160.53 (1 C), 155.65 (1 C), 153.82 (1 C), 148.06 (1 C), 145.75 (1 C), 141.11 (1 C), 140.56 (1 C), 140.23 (1 C), 138.93 (1 C), 136.96 (1 C), 135.32 (1 C), 121.56 (1 C), 121.17 (1 C), 120.76 (1 C), 118.56 (1 C), 118.21 (1 C), 117.78 (1 C), 114.91 (1 C), 108.33 (1 C), 105.13 (1 C), 104.87 (1 C), 52.67 (1 C), 51.24 (1 C), 50.67 (1 C), 44.84 (1 C), 29.78 (1 C), 28.47 (1 C), 27.63 (1 C), 26.97 (1 C), 25.34 (1 C), 25.03 (1 C), 22.63 (1 C), 21.83 (1 C) 15.65 (1 C), 13.07 (1 C).

HRMS: calculated for C$_{38}$H$_{45}$N$_2$O (M + H$^+$) 546.3560 m/z; measured 546.4042 m/z (M + H$^+$).

All obtained products were characterized by $^1$H and $^{13}$C NMR spectra on a Bruker FT-NMR Avance 400 (Ettlingen, Germany) spectrometer at 300 K, using TMS as internal standard and by high resolution mass spectrometry (HRMS) with a TRIPLETOF T5600 (ABSciex, USA) spectrometer. Fluorescence was characterized in a JASCO FP-8500 fluorescence spectrophotometer. To study the hydrolysis of sulfonic-Cy7Gal by β-Gal, 2 μl of human β-Gal enzyme (Biotechne) were added to PBS (pH 7) solutions of sulfonic-Cy7Gal (10$^{-5}$ M). Chromatograms of sulfonic-Cy7Gal, sulfonic-Cy7 and sulfonic-Cy7Gal+β-Gal after 30 min were obtained by reversed-phase liquid chromatography using a KromasilC18 column as the stationary phase, a mixture water-acetonitrile under gradient conditions (flow: 0.8 ml/min, initial condition 90:10 vol. %, final condition 10:90 vol. %) as mobile phase and a photodiode array detector. To study the specificity and selectivity of the probe to β-Gal, fluorescence intensity of solutions of sulfonic-Cy7Gal (20 μM, PBS, pH 7.4) was measured in the presence of cations (150 μM), anions (150 μM), small peptides (150 μM), proteins (150 μg/ml) and enzymes (150 μg/ml) at 37 °C after 30 min in a JASCO FP-8500 fluorescence spectrophotometer (λ$_{ex}$ = 580 nm; λ$_{em}$ = 665 nm).

## Cell culture, immunocytochemistry and imaging

The 4T1 mouse mammary tumor cell line was obtained from the American Type Culture Collection (ATCC, CRL-2539), cultured in DMEM supplemented with 10% FBS (Sigma) and penicillin-streptomycin and maintained in 5% CO$_2$ at 37 °C. For senescence induction, 4T1 cells were treated with 5 μM palbociclib (Selleckchem, pd-0332991) for 7 days. Cellular senescence was assessed by the senescence β-Galactosidase KIT acquired from Cell Signaling (#9860). 4T1 control and senescent cells were seeded in a clear flat bottom 96-well plate at a density of 3000 and 4000 cells per well, respectively.

After 24 h, cells were incubated with sulfonic-Cy7Gal (20 μM in DMEM with 10% FBS and 0.1% DMSO). For competition experiments with D-Galactose, a total of 200,000 control or 500,000 senescent 4T1 cells were seeded per well in a clear glass 6-well plate. The following day, cells were pre-incubated or not with D-Galactose (5 mM) for 30 min and then treated with sulfonic-Cy7Gal. Confocal images were acquired after 2 h in a Leica TCS SP8 AOBS confocal microscope (λ$_{exc}$ = 552 nm; λ$_{em}$ = 574-765 nm) and analyzed with the Image J software. For the in vitro cytotoxicity studies, 4T1 cells were plated in 96 well plates (4,000 control and 6000 senescent cells per well) and allowed to adhere. At 24 h post-seeding, the cells were incubated with varying concentrations of the sulfonic-Cy7Gal probe (diluted in DMEM) for 24 h. The cell viability was evaluated using the cellTiter-Glo® Luminescent assay (Promega). Luminescence was collected in a VICTOR Multilabel Plate Reader (Pelkin Elmer). Human endothelial cells (hUVECs) were obtained from EndoGRO™ (Sigma-Aldrich, SCCE001), cultured in a specific medium (EndoGRO-LS supplemented medium from Sigma-Aldrich) and maintained at 37 °C and 5% CO$_2$. For experimental set-ups, cells were seeded in 60 mm cell culture plates at a density of 500,000 cells per plate for pharmacological induction of senescence and half for control (untreated) cells. Senescence induction was carried out by treating the cells with 1 μM palbociclib for 7 days. For the analysis of WOS-Cy7Gal in live hUVECs, senescent and control cells were seeded in a specific chamber for optical imaging (8-well Ibidi® chamber) at a density of 20,000 and 10,000 cells, respectively. Once the cells were adhered to the plate, nuclei were labeled with 10 μg/ml Hoechst 33342 (Invitrogen) for 30 min. After careful washing of the cells, 20 μM of WOS-Cy7Gal in DMEM was added and images were obtained with an Olympus FV10i confocal microscope (using a 60x objective) within the first 10 min of incubation with the probe. The same procedure was followed using WOS-Cy7Ga and Sulfonic-Cy7gall in hUVECs entering replicative senescence after being passaged 29 times. For traditional SA-β-Gal activity detection, the Sigma-Aldrich kit for X-Gal histochemical staining was used. To detect by immunofluorescence senescence-associated markers, control and palbociclib-treated hUVECs were seeded in 48-well plates at a density of 12,500 and 25,000 cells/cm$^2$, respectively, on round glass coverslips. After 24 h, the cells were fixed with 2% paraformaldehyde (PFA) for 15 min and washed thoroughly with 0.1 M PBS. Prior to antigen-specific detection with primary antibodies, potential non-specific sites were blocked by incubating the cells in a blocking buffer (10% horse serum and 0.2% Triton™ X-100 in PBS) for 1 h at RT. Cells were then incubated with primary antibodies to Ki67 (Abcam, Ab15580; diluted 1:50), p21 (Abcam, Ab109520; diluted 1:100), lamin B1 (Abcam, Ab16048; diluted 1:100) and γH2AX (Millipore, 05-636-I; diluted 1:100) overnight at 4 °C. After several washes, cells were labeled with fluorescent secondary antibodies (Molecular Probes, diluted 1:600) for 1 h at RT. Finally, nuclei were stained with 4′,6-diamidino-2-phenylindole (DAPI, 1 μg/ml in distilled water) for 4 min and the glass coverslips containing labeled cells were mounted with FlourSave™ reagent (Calbiochem, 345789) for image acquisition with an Olympus FV10i confocal microscope. Quantitative analysis was performed using Image J software. Similarly, senescent and control hUVECs were fixed with 2% PFA for co-localization of p16 (Abcam, ab211542; diluted 1:100) and lamin B1 with WOS-Cy7Gal in an 8-well Ibidi® chamber. The probe was added at 20 μM in DMEM and incubated with the immunostained cells for 10 min at 37 °C. After washing the cells, images were immediately captured using an Olympus FV10i confocal microscope. To analyze and compare the signal patterns of the WOS-Cy7Gal and the sulfonic-Cy7Gal probes in hUVECs after 29 passages, we used a systematic approach using FIJI software. Roughly, we segmented the probe signal present in each cell after a background removal operation applied to all acquired fields. To reduce the contribution of the labeled area and focus only on the signal pattern, we skeletonized the obtained binary masks to erode and shrink them to their minimum pixel core. We then

quantified the total length of the resulting dots and branches, normalized to the corresponding cytoplasmic area, as a metric to compare differences in signal display patterns. The total normalized branch length exhibited by WOS-Cy7Gal was lower than that of sulfonic-Cy7Gal, consistent with the characteristic dotted pattern exhibited by this probe in cell lysosomes compared to the more filamentous signal found with sulfonic-Cy7Gal. To obtain the individual cytoplasm masks, first nuclei from the DAPI channel were segmented with a StarDist[60] custom model trained in a ZeroCostDL4Mic[61] notebook. Next, cell segmentation was conducted with the CellPose[62] 'CP_tissuenet' model in ZeroCostDL4Mic. Slightly dilated (maximum filter radius = 3 pixels) nuclei masks were subtracted (XOR) from the cell labels to generate the individual cytoplasm instances. Probe channels were preprocessed to subtract background (rolling ball= 5 pixels) and thresholder with the Otsu algorithm[63]. For each cytoplasm instance, the corresponding probe mask was skeletonized and analyzed with the 'Analyze skeleton 2D' plugin to measure the overall branch length. Lengths were normalized by cytoplasm area. For the analysis, we included a minimum of 30 cells for each probe.

### Knock-down experiments

For transient downregulation of *Glb1*, a total of 20,000 control or 50,000 senescent 4T1 cells were seeded *per* well in a 24-well plate. After 24 h, cells were transfected with TriFECTa® Kit DsiRNA Duplex siRNAs hs.Ri.GLB1.13.3 or scrambled (Integrated DNA Technologies, IDT), using Lipofectamine RNAiMAX reagent (Thermo Fisher Scientific) according to the manufacturer's instructions. The sequences for the IDT predesigned siRNA for GLB1 were sense: 5' rGrUrUrUrCrUrUr-ArArCrCrUrGrGrArCrUrCrUrCrArGrUrUrCCA 3'; and antisense: 5' rUrGr-GrArArCrUrGrArGrArGrUrCrArGrGrUrUrArGrArArArCrCrA 3'. At 48 h after transfection, cells were washed with PBS and fixed for 10 min with SA-β-Gal staining kit fixative. Then, cells were stained overnight using the SA-β-Gal staining kit (Cell Signaling, #9860) following manufacturer's instructions. The following day, cells were thoroughly washed with PBS and imaged using a colored bright field microscope. Other knock-down cells were treated with 20 μM sulfonic-Cy7Gal and imaged as before.

### Immunoblot

To determine the levels of β-Gal protein in control and senescent 4T1 cells, whole-cell extracts were obtained by using lysis buffer (25 mM Tris-HCl pH 7.4, 1 mM EDTA, 1% SDS, plus protease and phosphatase inhibitors). Cell lysates were resolved by SDS-PAGE, transferred to nitrocellulose membranes, blocked with 5% nonfat milk and incubated overnight with antibodies to β-Gal (Cell Signaling, #27198, diluted 1:1000) and to GAPDH (Cell Signaling, #2118, diluted 1:3000) as a loading control. Then, membranes were washed and probed with secondary antibodies conjugated to horseradish peroxidase that were detected with an enhanced chemiluminescence detection kit (Amersham Pharmacia Biotech).

### Mouse strains, treatments, IVIS® imaging and fluorometer measurements

BALB/cByJ mice were acquired from Charles River laboratories (France) and maintained at the Centro de Investigación Príncipe Felipe (CIPF). SAMR1 and SAMP8 mice, as well as C57BL/6 mice, were housed at the Universitat de València (UV). All mice were bred and housed, under 12 h periods of light/darkness, at a temperature of 20-22 °C with 40-60% humidity, and free accessible diet of pellets and water following European Union 2010/63/UE and Spanish RD-53/2013 guidelines and under official veterinary supervision. All animal procedures were approved by the CIPF and UV Ethics Committees for Research and Animal Welfare (CEBA) and conducted in accordance with the recommendations of the Federation of European Laboratory Animal Science Associations (FELASA). For senolytic treatment, a combination

of dasatinib (D) and quercetin (Q) was orally administered at different regimens to each mouse model. D (5 mg/kg) and Q (50 mg/ml) were dissolved with 20% PEG-n400 and 0.9% sodium chloride. 15-m C57BL/6 male mice received a 5-week treatment cycle with 2 doses *per* week and were tested for sulfonic-Cy7Gal and behavioral monitoring 15 days after the end of the treatment. On the other hand, 7-m SAMP8 male mice received the same drug mixture for consecutive 5 weeks, alternating a regime of 5 or 1 administrations *per* week. In this case, sulfonic-Cy7Gal and behavior monitoring were evaluated in the first 21 and 58 days after the end of treatment. The same cohort of animals received another round of treatment at 10 months of age and were tested only within the first 21 days after the completion of this second cycle of treatment. Breast 4T1 tumors were established by using 4T1 cells in BALB/cByJ mice. Cells were trypsinized, counted with a LUNA™ Automated Cell Counter, and $0.5 \times 10^6$ cells in a volume of 100 μL were injected subcutaneously in the left mammary fat pad of 28- to 34-week-old BALB/cByJ female mice. Palbociclib dissolved in 50 mM sodium lactate (pH 4) was administered at different doses by daily oral gavage for 7 days. Tumor volume was measured every two days with a caliper and calculated as $V = (a \times b^2)/2$, where a is the longest and b is the shortest of two perpendicular diameters. Mice were anesthetized by inhalation of 2% isoflurane and intraperitoneally (i.p.) injected with sulfonic-Cy7Gal (23.3 mg/mL; 100 μL, 2.5 μmol *per* mice) or WOS-Cy7Gal (19.7 mg/mL; 100 μL, 2.5 μmol *per* mice) and maintained in an IVIS® Spectrum imaging system (PerkinElmer) for 15 min taking photographs every 2 min ($\lambda_{exc}$ = 535 nm; $\lambda_{em}$ = 640 nm; exposure time: 10 s). Urine was collected after mice recovered from anesthesia in an Eppendorf tube and analyzed directly by IVIS® ($\lambda_{exc}$ = 535 nm; $\lambda_{em}$ = 640 nm; exposure time: 1 s). Urine fluorescence was analyzed with a fluorometer (JASCO FP-8500). For this analysis, 5 μL of urine was diluted in 95 μl of distilled water and fluorescence was recorded at ($\lambda_{exc}$ = 535 nm; $\lambda_{em}$ = 560 nm). The amount of sulfonic-Cy7 fluorophore excreted in urine was calculated through a calibration curve. To obtain the calibration curve, a stock solution of sulfonic-Cy7 in blank urine from mice was prepared. Serial dilutions were prepared with the same urine and 5 μL of each sulfonic-Cy7 urine solution was added to 95 μl of distilled water and measured in the fluorometer under the same conditions. In order to assess the biodistribution of the fluorophores from the sulfonic-Cy7Gal and WOS-Cy7Gal probes, fluorescence was measured in urine and plasma. Plasma samples were obtained 10 min after the probe or vehicle injection by submandibular puncture of mice with a 25 G needle. Blood was collected in heparinized tubes and, subsequently, these samples were centrifuged at 350 g for 4 min and the supernatant (plasma) was collected. From each plasma sample 10 μL was diluted in 90 μL of distilled water and fluorescence reading was performed in the fluorometer ($\lambda_{exc}$ = 535 nm; $\lambda_{em}$ = 560 nm). For the calibration curve, a stock solution of sulfonic-Cy7 in blank plasma from mice was prepared. Serial dilutions were prepared in the same plasma and 10 μL of each sulfonic-Cy7 urine solution was added to 90 μL of distilled water and measured in the fluorimeter under the same condition. Mice were immediately euthanized after collecting the urine by $CO_2$ exposure, and tumors and organs (lungs, liver, kidney, spleen, brain and bladder) were immediately harvested and freshly analyzed by IVIS® ($\lambda_{exc}$ = 535 nm; $\lambda_{em}$ = 640 nm; exposure time: 1 s). IVIS® images were analyzed using the Live Imaging software from Caliper Life Sciences. The fluorescence quantification associated with these images involved defining regions of interest (ROIs) to outline the fluorescence signals, measuring the average radiant efficiency related to each ROI, and subtracting the background according to the IVIS® Spectrum user's manual. On the other hand, when analyzing mice of different age and strain, the dose of sulfonic-Cy7Gal administered was adjusted to the body weight of each animal (140 mg/kg, 200 Ll i.p.). After injection, all animals were anesthetized with isoflurane (2%) for 15 min and, upon awakening, urine samples were collected individually. Because the volume of

micturition may vary between young and old animals, we measured in a fluorometer (Horiba Scientific Fluoromax-4) the counts *per* second (CPS or arbitrary units) of a 1/20 dilution in distilled water of 5 μL of the collected urine and the resulting units were multiplied by the total volume of urine collected in each sample. Notably, for age-related studies and longitudinal senolytic setups, we used male mice to avoid sex variability on aging and micturition[64,65].

## HPLC-MS

HPLC-MS measurements were obtained using an eluent gradient method from $H_2O$-methanol (100:0 v/v) to $H_2O$-methanol (0:100 v/v) at 10 min with a flow rate of 0.5 Ll/min with a kromasil C18 column. Mass spectroscopy chromatograms were recorded with an Agilent Ultivo mass spectrometer equipped with a triple Q-TOF detector using a dual selected ion monitoring (SIM) function at 1034 m/z and 705 m/z simultaneously, corresponding to the sulfonic-Cy7Gal and sulfonic-Cy7 fluorophore, respectively. The calibration curve was obtained by measuring known concentrations of the sulfonic-Cy7Gal probe and the fluorophore in $H_2O$ using the dual SIM method. The amount of sulfonic-Cy7Gal and sulfonic-Cy7 fluorophore excreted was calculated by measuring 5 μL of urine sample diluted in 500 μL $H_2O$ and multiplying the corresponding concentration, obtained with the calibration curve, by the volume of urine collected for each sample. The percentage of probe or dye excreted in the urine was obtained by relating it to the amount of sulfonic-Cy7Gal injected.

## Histology

Tumors were fixed with 4% PFA in PBS for 4 h, embedded in paraffin, sectioned at 5 μm and mounted onto pre-coated slides. Sections were dewaxed, incubated with the primary antibody Ki67 (Cell Signaling, #9129, diluted 1:800) and then with the corresponding peroxidase-conjugated secondary antibody in an automated immunostaining platform (Leica Microsystems Bond RXm). After the peroxidase detection, the sections were counterstained with hematoxylin, dehydrated, and coverslipped. In some animals, tumors together with kidneys and liver were extracted at the endpoint and snap-frozen in liquid nitrogen. Next, the samples were embedded in OCT and sectioned with a cryostat at 7 μm thickness. Sections were used for the detection of SA-β-Gal with the X-Gal histochemical reaction as previously reported[55].

## Cytometry

For ex vivo assessment of senescence burden, 7-m SAMP8 and SAMR1 mice were euthanized by cervical dislocation and the right kidney and a piece of the right lobe of the liver were extracted. These tissues were subjected to enzymatic digestion with a mixture of 1 mg/ml collagenase/dispase (Roche) and 0.2 mg/Ll DNAse (Labclinics) diluted in 2 ml RPMI *per* sample. Tissue enzymatic dissociation was performed using a gentleMACS™ Octo Dissociator with heaters (Miltenyi). Next, the digested tissue pieces were mechanically dissociated and filtered through a 40 μm nylon filter to obtain a single cell suspension. In order to avoid the auto-fluorescence noise of some cells and to target cell populations that are found in many organs, cells were incubated with CD31 and CD45 antibodies (BD-Bioscience, 740239 and 563890, respectively; both diluted 1:100) to label endothelial and immune cells. Thus, the results obtained are the product of the analysis of senescence markers in both populations. For SA-β-Gal detection, a proportion of live cells was incubated with 20 μM of WOS-Cy7Gal in FACS buffer (10% Hank's Balance Salt Solution, 10% EDTA, 1% HEPES, sterile $H_2O$) for 15 min prior to flow cytometry assessment (BD LSR-Fortessa cytometer) together with 0.1 μg/ml DAPI to exclude dead cells. The remaining cells were fixed and permeabilized using BD Pharmingen™ Transcription-Factor Buffer Set for detection of intranuclear senescent markers. Cells were first incubated with the primary rabbit antibodies to p16 (Abcam, ab211542; diluted 1:100) or lamin B1 (Abcam, ab16048; diluted 1:100) and then with secondary anti-rabbit antibodies

labeled with Alexa Fluor™ (Invitrogen; diluted 1:400) and analyzed in LSR-Fortessa cytometer. In a similar vein, for the in vitro assessment of senescence in hUVECs, senescent and control cells were incubated with 20 μM WOS-Cy7Gal (in DMEM) for 15 min before evaluating the MFI by flow cytometry. Additionally, to detect other senescence markers in combination with this probe, cells were permeabilized and fixed, using the same kit as mentioned previously, for subsequent immunostaining to p16 and lamin B1 antibodies. Following incubation of cells with secondary antibodies, 20 μM WOS-Cy7Gal was added for 10 min. The cells were then centrifuged (500 x*g*, 5 min) and resuspended in FACS buffer, and the fluorescence levels were immediately evaluated in the LSR-Fortessa flow cytometer. Flow cytometry analysis was carried out with FlowJo_v10.8.1 software.

## Behavioral tests

Anxious behavior and general locomotor activity were measured in the open field test, using a gray squared Plexiglas open box (50 cm²) (Panlab, S.L.). Briefly, mice were gently introduced in the open field and videotaped for 10 min in a dim light room to which they have been previously habituated. Periphery/center areas were automatically established by the Smart video tracking software (Panlab, Harvard Apparatus) and the percentage of time spent in the central area was calculated for each mouse. Trajectory of movement and motor behavior were also analyzed. Anxious behavior was also evaluated in an elevated plus maze as described previously[66]. Briefly, mice were placed for 5 min in an elevated gray Plexiglas cross-shaped maze (30 cm L each arm x 5 cm W x 40 cm H) with two closed arms, two open arms, and a center area. Behavior of the mice during free exploration of the maze was recorded using a video tracking system and the percentage of time spent in the open arms calculated. Anxiety is reflected as a higher avoidance of heights/open spaces.

## Statistical analyses

All the statistical analyses were conducted in GraphPad 5.0 (Prism) software. All the sample sizes and statistical tests are specified in the figure legends. The Rout method (Q = 5%) was used to identify and exclude outliers. Unpaired or paired t-tests and one-way or two-way ANOVA were used when appropriate. All reported p-values are based on two-tailed tests and results include a 95% confidence interval. For each animal experiment, groups were established before tumorigenesis or treatment with sulfonic-Cy7Gal or WOS-Cy7Gal, and therefore no randomization was used in the allocation of groups. Investigators were not blinded to the groups and treatments during the experiments, with the exception of the probe read-out and behavioral tests performance and analysis.

## Reporting summary

Further information on research design is available in the Nature Portfolio Reporting Summary linked to this article.

## Data availability

The authors declare that the data supporting the findings of this study are available within the paper and its supplementary information files. Moreover, source data is provided as an Excel file with this paper. Source data are provided with this paper.

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

## Acknowledgements

We acknowledge the support of the Servicio Central de Soporte a la Investigación Experimental of the Universitat de València (SCSIE-UVEG). We are grateful to Biorender.com for the illustrations of mice used in the temporal scheme of the senolytic treatment in Figs. 5 and 6 (agreement number: IX264V4WHR). We thank Dr. D. Muñoz-Espín for providing us with siRNAs for knock-down experiments and Dr. S. Gil for his assistance in NMR analysis. Work in the laboratory of R. M.-M was supported by grants from the Spanish Government (PID2021-126304OB-C41) and the Generalitat Valenciana (PROMETEO CIPROM/2021/007). Work in the laboratory of I.F. was supported by grants from the Spanish Government (PID2020-117937GB-I00, RED2018-102723-T, and CIBERNED CB06/05/0086) and Generalitat Valenciana (PROMETEO/2021/028). We also acknowledge the financial support from the Spanish Government (SAF2017-84689-R project MINECO/AEI/FEDER, UE) and the Generalitat Valenciana (project PROMETEO/2019/065) to M. O. A. P. V. was also founded by CITSAM (2021-283-001). S. R.-V. (FPU 16/03714) and B. L.-T. (FPU15/02707) were recipients of FPU predoctoral contracts from the Spanish Ministerio de Educación; P. M-R. was the recipient of a pre-doctoral contract (LCF/BQ/DE18/11670015) from Fundació la Caixa; J. F. B. was the recipient of a postdoctoral contract (PAID-10-17) of the Universitat Politècnica de València.

## Author contributions

S. R.-V., B. L.-T., J. F. B., F. S., I. F., and R. M.-M. conceived and designed the research. B. L.-T., M. D., and J. F. B. synthesized and characterized all organic molecules. J. F. B. performed HPLC studies. S. R.-V. and B. L.-T. carried out all in vitro and in vivo studies and analyzed the data. A. G.-F., I. G., and M. J. P. assisted with the in vitro and in vivo experiments. M. O. helped in designing the in vivo experiment in the chemotherapy-induced 4T1 breast cancer model. S. R.-V. and A. P.-V. carried out the behavioral analyses, S. R.-V. and P. M.-R. performed the cytometry studies. I.F. and R. M.-M. wrote the manuscript with feedback from all the authors.

## Competing interests

B. L.-T., A. G.-F., I. G., M. O., R. M.-M., F. S, and J. F. B. have filed a patent application related to this research with the "Oficina Española de Patentes y Marcas". R.M.-M. is co-founder of Senolytic Therapeutics, Inc. (USA) and Senolytic Therapeutics, S.L. (Spain). The remaining authors declare no competing interests.
