## [Peer Review File · Nature Communications]

REVIEWER COMMENTS

Reviewer #4 (Remarks to the Author):

Dear authors

I was asked to specifically review the answers to reviewer 3, which I did. I have therefore pasted my comments under this review. As I did this in the original file, I have uploaded it separately as attachment, as I think this will be more comprehensible.

I have carefully read and considered all adapted changes in view of the comments. In my opinion you have improved the manuscript according to the comments raised. Although I do have some minor suggestions that may further improve the manuscript, I do not see major concerns. As an independent additional reviewer I have the difficult task to not only judge the work of my peers, but also the work of a peer reviewer. Therefore I would like to ask you to carefully consider the minor comments I have, and address them appropriately, where possible. It is not my intention to impose more work, but I do think that my suggestions may further improve the manuscript for publication in Nature Communications, and some also agree with earlier comments raised.

Ref: NCOMMS-23-37044-T

A renal clearable fluorogenic probe for the in vivo detection of β -galactosidase activity during aging and senolysis

Authors' response to reviewers' comments:

Provided below is our detailed response, addressing the questions and comments raised by Reviewer #3 (*verbatim*, in bold) during the most recent round of revisions at *Nature Aging*. This accompanies a new revised version of our manuscript that accommodates the changes, highlighted in blue.

Reviewer #3. Remarks to the Author:

In this disappointingly revised manuscript, the authors have made a feeble attempt to address a few of the issues raised by reviewers. While there may be some marginal improvement in the quality of this manuscript after revision, it remains plagued by major concerns that undermine its scientific impact and novelty.

We are sorry but we cannot agree with the comment of the reviewer in relation to the "*feeble attempt to address a few of the issues raised by reviewers*". We spent months of work to demonstrate that the probe can be used in different aging models and in longitudinal studies to monitor a senolytic treatment, following suggestions of Reviewers #1 and #2 and the Editor.

I have carefully read and considered all adapted changes in view of the comments. In my opinion the authors have improved their manuscript according to the comments raised. Although I do have some minor suggestions that may further improve the manuscript, I do not see major concerns. As an independent additional reviewer I have the difficult task to not only judge the work of my peers, but also the work of a peer reviewer. Therefore I would like to ask the authors to carefully consider the minor comments I have, and address them appropriately, where possible. It is not my intention to impose more work, but I do think that my suggestions may further improve the manuscript for publication in *Nature Communications*, and some also agree with earlier comments raised.

Major concerns.

The authors have audaciously claimed that no method or reported probe exists for in vivo detection during aging or after senolytic interventions, conveniently ignoring the existence of well-documented references that have developed β -Gal-based probes for precisely that purpose (Nat Aging, 2023, 3(3):297-312; J Med Chem, 2021, 64(24):17969-17978; Anal Chem, 2020, 92(18):12613-12621).

We thank Reviewer #3 for his/her comments. However, note that we did not attempt to claim that "*no method or reported probe exists for in vivo detection during aging or after senolytic interventions*". In the Introduction of our manuscript, we indicated: "*However, new strategies aimed at the detection of senescent cells and their monitoring during aging or after senolytic interventions are still required. Although the quantification of cell senescence in biopsies from different human tissues is an active area of research, non-invasive procedures to monitor cellular senescence in vivo have not yet been developed*". Note that the second sentence refers to humans, not to animal models. Our assertion does not suggest the absence of reported probes for in vivo β -Gal detection. Instead, it pertains to the absence of probes that can be retrieved from urine. In fact, the provided citations by the reviewer are good examples of probes that can be detected *in vivo* (in animal model), but not recovered in biological fluids.

- *Nat Aging*, 2023, 3(3):297-312: This paper describes a photoactivatable senolytic approach based on a system that is an enzyme substrate of senescence-associated β -galactosidase (SA- β -gal).
- *J Med Chem*, 2021, 64(24):17969-17978: This paper described a probe that can be used in vivo non-invasively, similar to our WOS-Cy7Gal, but it is not excreted in urine.
- *Anal Chem*, 2020, 92(18):12613-12621: The same as the previous one.

We have provided a clearer explanation of the distinction in the updated Introduction (see highlighted text) and added the two last references suggested by the reviewer.

Your explanation here is clear. However, I was a bit confused by the highlighted added text. Especially the part 'the signal detection relies on imaging methods'. Maybe also because it depends on what one considers non-invasive; some fins injection invasive, others do not even find imaging invasive, unless it is combined with an operation.

In my opinion, what you want to point out here is that there are certainly imaging methods available that make use of probes for senescence, but that you now developed a probe that can also be used without imaging modalities; by measuring it in collected urine. Measuring it in urine is an extra benefit. However if you only measure it in urine, you will not be able to pinpoint which tissue is most affected. As you can also use the probe for whole body imaging, and pinpointing the signal to specific tissues, as you show in the manuscript, I would suggest you highlight both these aspects. As you explain it well above, maybe you can use some of this text to rephrase the highlighted text?

Furthermore, the authors fail to acknowledge that β -Gal is far from a specific biomarker for cellular senescence, and the presence of β -Gal activity during aging or senolytic therapy does not necessarily indicate the presence of senescent cells.

This has been a key point in the revisions. Reviewer #2 already indicated this in his/her second report: "*describe what we are actually measuring is β -galactosidase activity- which could be associated with senescence*". Following that indication of Reviewer #2, we eliminated from the manuscript any reference to senescence-associated β -Gal activity and to senescent cell burden and indicated along the text that our probe measures β -Gal activity and that the probe readout can be correlated with changes during aging and senolytic interventions that also result in parallel changes in anxious behavior. We even changed the title of the manuscript to: "A renal clearable fluorogenic probe for the in vivo detection of β -galactosidase activity during aging and senolysis". Therefore, note that in this last version we are not claiming in the manuscript that β -Gal is a specific biomarker of cellular senescence.

This comment was addressed properly

To add insult to injury, they attempt to tout the molecular scaffold of sulfo-Cyanine7 as a supposedly innovative near-infrared (NIR) fluorophore with improved water solubility, conveniently ignoring the fact that it is nothing more than a widely used and unremarkable component. As a result, this work contributes very little to the field of senescence detection or the design of molecular probes, rendering it virtually insignificant.

We never claimed in our manuscript that the molecular scaffold of sulfo-Cyanine7 is a supposedly innovative near-infrared (NIR) fluorophore, neither we claimed that sulfo-Cyanine7 has an improved water solubility. The presence of the sulfonic groups is to make the probe more diffusible and renally clearable, making the Cy7-Gal the first probe to detect β -Gal activity in biological fluids. As we mention in the text: "*Cy7-Gal exhibited a more diffuse cellular pattern that was compatible with lower retention in lysosomes*". Our probe is excreted and gives an organismal read-out of β -Gal activity in urine that correlates with age progression and is able to monitor the effects of senolytic treatment. In our perspective, this represents an advancement within the field of aging research. Note that this is an innovative approach and very few

examples of urine-recovered probes have been reported, none of them for the *in vivo* monitoring of β -Gal activity. Moreover, note also that these renal clearable probes are closer to the potential translation into humans than other probes, which can be used also in a non-invasive manner, but require imaging.

Although not a central point of our manuscript, note that most available probes are mainly eliminated by the reticuloendothelial system, which can lead to their accumulation in the liver and spleen with the consequent potential organ toxicity and side effects. An alternative to this is the design of renal clearable probes which is an area of growing interest. Moreover, additionally in our case the probe, in an OFF state, is specifically transformed to an ON state by the action of β -Gal activity in cells/tissues *in vivo*, has a rapid elimination by the kidney into the urine where it can be easily measured.

In my opinion you have addressed this comment properly. I have to admit, upon first read of the manuscript, it was not entirely clear to me why first these sulfonic groups were omitted at first. If you add this sentence : 'The presence of the sulfonic groups is to make the probe more diffusible and renally clearable, making the Cy7-Gal the first probe to detect β -Gal activity in biological fluids.' either in the abstract or at the end of the introduction, this would make it more clear.

Given these glaring shortcomings, I strongly advise against the acceptance of this manuscript in Nature Aging. At best, it may be suitable for publication in a more specialized journal, provided the authors diligently address the multitude of issues that still plague their work.

Major technical issues.

1. In this revised manuscript, the authors describe their probe to detect β -Gal activity during aging and senolysis, instead of senescence-associated β -Gal activity. Thus, the authors should improve the specificity of the probe for the detection of senescence.

Please see our comment above and previous comments by Reviewer #2.

Our probe is able to monitor β -Gal activity *in vivo* and its read-outs correlate with age progression and the effects of senolytic treatment. We agree with the reviewer that it would be interesting to develop a probe that detects senescence-associated β -Gal activity. However, this is not an easy task (and is out of the scope of our manuscript). The reason is that SA- β -Gal activity is not due to a specific β -Gal enzyme or a different lysosomal pH, but a higher activity resulting from increased lysosomal mass and increased expression of GLB1, the gene encoding the lysosomal β -Gal enzyme (Kurz et al., 2000, *J. Cell Sci.* 113:3613; Lee et al., 2006, *Aging Cell* 5:187). Therefore, a probe specifically generated to detect senescence is difficult to achieve based alone on β -Gal activity. In fact, most of the probes reported by now to detect senescence rely on the detection of the overexpressed lysosomal β -Gal enzyme in senescent cells (named as SA- β -Gal activity).

In my opinion this comment is properly addressed. I think this is a good point of discussion, and this should be added as such to the discussion section, as well as how you envision the probe to be used.

2. In Fig 1b, a few senescent hUVECs have two Hoechst-labeled nuclei in one cell, which means that they still replicate. In this case, why does the fluorescence of WOS-Cy7Gal turn on in these cells?

We thank Reviewer #3 for this appreciation. It is true that a few hUVECs entering palbociclib-induced senescence are binucleated. It can also be seen in the X-Gal staining of hUVECs undergoing replicative senescence (Extended Data). To use another source of data unrelated to our work, the web page of the Cell Signalling company uses a picture of β -Gal histochemistry to advertise their Senescence β -Galactosidase Staining Kit that also stains binucleated cells

(<https://www.cellsignal.com/products/cellular-assay-kits/senescence-b-galactosidase-staining-kit/9860>). Binucleation or tetraploidy usually reflects endomitosis, i.e. cells enter M phase but exit prematurely, before completing cell division. Binucleated cells are found at very low frequency in healthy primary cell cultures, but their proportion can be increased depending on culture conditions. In any case, this cell state, as observed in our hUVEC cultures (hUVECs are primary cells), is not an indication that the cell is dividing but that it underwent endomitosis before entering palbociclib induced senescence. Interestingly, tetraploidy can also lead to senescence (Panopoulos et al., 2014, *Mol Biol Cell* 25:3105).

This is a valid explanation, and it would be very helpful if this would also be added in one or two sentences to the results discussion, as it does take away the question of why binucleated cells are present.

3. In Fig 1f, the immunostaining of p16 is not specific, which should be confined to nuclei and has no colocalization with nuclei. The authors should optimize the immunostaining of p16 and add the staining of nuclei.

Please see the type of expected staining as shown in the web page of the provider of the antibody used: <https://www.abcam.com/products/primary-antibodies/cdkn2ap16ink4a-antibody-epr20418-ab211542.html>. We can also provide citations in which a senescence-associated increase in the levels of p16 can be found in the cytoplasm and/or nucleus depending on the trigger (i.e., doxorubicin vs confluence). See, for example Spallarossa et al., 2010, *PLoS One* 5:e15583.

Specificity of p16 stainings has been an ongoing debate. Actually, only when you show no signal with the p16 antibody on p16ko tissue/cells would I be convinced that it is specific. However, in our lab we have used this antibody and it does seem to be specific as far as we can tell, at least for immunohistochemical tissue stainings. Most IF stainings I have seen show nuclear staining but some also show cytoplasmic staining. Therefore this is debatable. Would it be possible to add p21 staining as this should give a similar pattern as p16 and would support your data? Or do you have other stainings that would support the present p16 stainings? I do not want to impose more work, but it would be somehow good to be sure that we are looking at specific p16 staining.

4. In Fig 2a, the authors insist that the signals elsewhere other than in the bladder are background. In this case, the data processing should be optimized to minimize the background to increase the data accuracy. Moreover, representative pictures should be provided to account for the results.

We thank Reviewer #3 for his/her comment. Figure 2a illustrates the formulas of the compounds. Maybe the referee means to refer to Figure 2b. If this is so, this is a question already commented by the reviewer previously. In previous reports we included representative images for the reviewer that illustrate the signal obtained in whole animals. In this case, fluorescence can be detected at the site of the injection (intraperitoneal) and, in a palbociclib dose-dependent manner, in the tumors and the bladder (see Extended Data Figure 5a). Figure 2b shows representative pictures where fluorescence at the site of the injection is not shown and only emission in the lower belly of the animal (where the urinary tract is) is shown. Also note that all the data is coherent. Thus, as the hydrolyzed Cy7Gal probe is urinated it is expected that the in vivo emission in the animal (before mice urinate) comes from the bladder. Note that also the bladder has a large fluorescence ex vivo when the probe Cy7Gal, but not when the WOS-Cy7Gal probe is used (Figure 2f). This is coherent with the specific elimination of the hydrolyzed Cy7Gal probe in the urine. Nevertheless, and following the advice of the reviewer, we have revised Figure 2b and 3a, reducing the background of the IVIS® images to further limit the probe-associated signal and providing a representative image for each condition. Furthermore, we have delineated ROIs in the bladder area of BALB/cJyB mice bearing 4T1 tumors and treated

with different doses of palbociclib, and also delimited this zone in BALB/cJyB mice from young and old age groups. We have quantified the fluorescence within these regions, and the corresponding data is now included in Extended Data Figure 5. We hope this can prevent any potential confusion.

In my opinion this comment is properly addressed.

5. In Fig 4g,h and Fig 5 e,f, the coefficient of determination R2 is less than 0.5, which means there is a poor linear correlation between Cy7 fluorescence in urine and related factors.

Figure 4 illustrates our cytometry-based analysis and Figure 5e,f illustrate behavioral analysis in the elevated plus maze. We understand that the referee may be referring to Figure 5g,h and Figure 6e,f. Figure 5 g shows the significant linear correlation between Cy7 fluorescence in urine and the percentage of time spent in the center of the open field, in mice treated with D+Q or vehicle. Figure 5f shows the significant linear correlation between Cy7 fluorescence in urine and the percentage of time spent in the open arm of the EPM test in mice treated with D+Q or vehicle. Figure 6e shows the significant linear correlation found between the levels of Cy7 fluorescence in urine and anxious behaviour in mice treated with D+Q or vehicle 15 days after the treatment, whereas Figure 6f, shows that there is not a significant correlation between Cy7Gal and anxious behaviour when assessed in the long term after the senolytic intervention. Note that correlations with relatively low R2 coefficients are normal when using behavioural tests. The positive R2 coefficients found indicate a real relationship between the predictor (anxiety) and the response variable (Cy7 in urine) as indicated by the significance analysis.

6. The exponential powers of radiant efficiency between the IVIS imaging pictures and quantification data should not have such a big difference. For example, Fig. 2f (x108) and 2g (x1011), Fig. 3a (x106) and 3b (x109), Extended Data Figure 6 a (x107) and b (x109). The authors should double-check these data.

We have double-checked all the data.

I also do not understand this part; in Fig2g we see the quantification of 2f, where for the bladder we see a signal x10⁸, for the tumor x10⁹, and for the kidney x10⁷. How then can the radiation efficiency be expressed in x10¹¹ for all these tissues in 2g? Could you clarify this? Maybe there is some conversion calculation that I have missed.

7. In the revised manuscript, the author adopts an unusual form of figure representation, arranging bar charts and data dots separately. However, in Extended Data Figure 7, the “-Cy7Gal” group’s data dots were not fit with the bar chart. This bar chart’s height was much lower than the data dots shown in the figure. We require the author to provide the specific values for dots and bars in the current Extended Data Figure 7. The author should clarify how this situation happened. Did the figure generated by the software automatically or have been adjusted manually?

This is a form of figure representation which we understand that it is quite common (at least it is common for us). We would like to stress that we have not adjusted any graph manually. We have shown all the data points and the bars represent the mean value in all the graphs. We have listed

	Cy7Gal		wo. Cy7Gal
	2-m	14-m	14-m
BLADDER	0,01	2,67	0,04
	0,28	1,81	0,03
	0,57	0,97	0,03
		0,48	
	mean	0,2867	1,4825
SD	0,2801	0,9635	0,0058
SEM	0,1617	0,4818	0,0033
BRAIN	0,26	1,18	0,25
	0,63	1,44	0,23
	0,54	0,51	0,07
		1,14	
	mean	0,4767	1,0675
SD	0,1930	0,3947	0,0987
SEM	0,1114	0,1974	0,0570
LUNG	0,09	1,11	0,06
	0	0,63	0
	0,42	0,07	0,09
		0,37	
	mean	0,1700	0,5450
SD	0,2211	0,4407	0,0458
SEM	0,1277	0,2204	0,0265

the data in the table below. If the Editor deems it appropriate, we can include these specific values for points and bars in Extended Data Figure 7.

Normally one would show dotplots where the dots fall inside of the boxes. As an example:

In the figures of this manuscript all dots are outside of the boxes, this could be the source of confusion. My preference would be to have the dots inside the boxes. I leave it up to the editor to decide which is the right representation for the journal.

8. In Extended Data Figure 3a, the control 4T1 group showed no positive β -gal staining results. However, several reported articles focused on 4T1 tumor detection strategies were based on β -gal detection. For example, [Evolving an Ultra-Sensitive Near-Infrared β -Galactosidase Fluorescent Probe for Breast Cancer Imaging and Surgical Resection Navigation. *ACS sensors*, 2022, 7(12): 3829-3837]. We believed that senescent 4T1 may have higher β -Gal levels than the control 4T1, but it doesn't mean there should be no X-gal staining positive region in the control group. Actually, in Extended Data Figure 3b, the control 4T1 cells + Cy7Gal probe showed a higher signal than the background, proving that there should be certain β -gal in normal 4T1. Too absolute results may mislead readers to think that there is no β -gal expression in 4T1.

This is an interesting question. There are multiple reported studies with the X-gal kit showing no positive staining results in normal cells and a clear positive staining in senescent cells when using staining. We have found very similar X-gal staining in normal and senescent 4T1 cells in other studies (see for instance *Anal. Chem.* 2021, 93, 3052). Note that no staining does not mean that there is not β -Gal, but that the β -Gal level is not enough to display a colorimetric signal taking into account the special conditions in which the X-gal kit is used (suboptimal pH for the β -Gal enzyme; Debacq-Chainiaux et al., 2009, *Nature Protoc.* 4, 1798). In addition, to use another source of data unrelated to our work, in the web page of the Senescence β -Galactosidase Staining Kit #9860 from Cell Signalling Technologies there is a picture showing no staining at pH 6 on normal WI38 cells at population doubling 29 but a clear blue X-gal staining of senescent WI38 cells at population doubling 36. In addition, note that it is very difficult to compare the degree of senescence, in the absence of an inducer, between our work and the cited one as the latter does not inform about number of cells per graft or time between transplantation and analysis. One has to keep in mind, however, that 4T1 are immortal cells of a tumor cell line and they do not enter senescence unless forced to do so (i.e. with palbociclib).

This comment has been properly addressed

9. In Extended Data Figure 3e, the author tried to prove that the Cy7Gal probe had no cytotoxicity. However, at concentration 1.1 (log[Cy7Gal]) μ M, the viability of control 4T1 cells + Cy7Gal was only 60%, probably reaching the level of some chemotherapy drugs. The author should explain why this concentration was so unique because higher concentrations also showed no obvious cytotoxicity.

The data in Extended Data Figure 3 shows that the Cy7Gal probe exerts no detrimental effects on cell viability. As many as 10 increasing concentrations were tested. As mentioned by the reviewer, we found an apparent effect with one of the concentrations. We believe that the graph unmistakably illustrates the absence of cytotoxicity. Nonetheless, we have generated a new graph. We have considered that the way we presented the concentrations of the Cy7Gal probe in this graph was a bit confusing, as we represented them as $\log[\text{Cy7Gal}]$. We have simplified the graph to more clearly indicate the concentrations used in the experiments and much higher: 25, 50, 100, and 200 μM .

I have not seen the original figure, therefore it is difficult to judge this adaptation. I do have an additional question; cytotoxicity is very important as you would like to use this probe in vivo, and maybe even in humans. Now there is a high clearance rate so long-term toxicity of the probe may not be an issue; have you looked at this, or any gathered any data on this? For the viability assay you use one that measures ATP. ATP content can be different between senescent cells and non-senescent cells. Have you measured cytotoxicity in any other manner, such as simply counting cells, or another type of viability assay with another outcome measure? I ask this as now in your graph the senescent cells seem more viable, which seems unlikely and may be due to the difference in ATP content. As you especially emphasize the non-invasiveness of the probe, any data you have on (non) cytotoxicity of the probe would be very helpful.

10. In Extended Data Figure 3f caption, the author declared that “Note that while WOS-Cy7gal displayed a dotted pattern, Cy7Gal showed a more diffuse one”. Considering the two pictures were quite similar, we suggested the author use Image J to analyze the pictures. On a linear path, a “diffuse” pattern should be smoother than a “dotted” pattern.

We thank the reviewer for the suggestion. We have now applied image analysis to demonstrate that the pattern of cell staining is, indeed, different. We have added this information as new Extended Data Figure 2f,g and a paragraph in Methods explaining how the image analysis was done.

This comment has been properly addressed

11. In Extended Data Figure 4, mice carrying orthotopic 4T1 were treated with Palbociclib. 50 mg/kg or 100 mg/kg Palbociclib treatment led to a 50% tumor shrinkage on day 14. However, in Extended Data Figure 6a, mice treated with 50 mg/kg Palbociclib showed an even larger tumor size than the untreated group, whereas the 100 mg/kg Palbociclib treatment group had a smaller tumor size. The author should provide detailed information about when the IVIS experiment, the Ki67 staining and β -Gal staining were carried out in the Methods section. In addition, the author needs to confirm the tumor size in the IVIS image had no controversy when compared with the tumor growth curves.

We do not see any controversy between the tumor size in the IVIS image when compared with the tumor growth curves. Please note that as for tumor size, Extended Data Figure 4 shows measures with a caliper, whereas Extended Data Figure 6a shows dissected tumors and, therefore, the sizes may not be accurate due to resection effects.

This comment has been properly addressed

Ref: NCOMMS-23-37044-T

A renal clearable fluorogenic probe for the in vivo detection of β -galactosidase activity during aging and senolysis

Authors' response to reviewers' comments:

Provided below is our detailed response (in brown), addressing the questions and comments raised by Reviewer #4 (in blue) in relation to our answers to Reviewer #3 comments (in bold). This accompanies a new revised version of our manuscript that accommodates the new changes, highlighted in blue.

Reviewer #3. Remarks to the Author:

In this disappointingly revised manuscript, the authors have made a feeble attempt to address a few of the issues raised by reviewers. While there may be some marginal improvement in the quality of this manuscript after revision, it remains plagued by major concerns that undermine its scientific impact and novelty.

We are sorry but we cannot agree with the comment of the reviewer in relation to the “feeble attempt to address a few of the issues raised by reviewers”. We spent months of work to demonstrate that the probe can be used in different aging models and in longitudinal studies to monitor a senolytic treatment, following suggestions of Reviewers #1 and #2 and the Editor.

I have carefully read and considered all adapted changes in view of the comments. In my opinion the authors have improved their manuscript according to the comments raised. Although I do have some minor suggestions that may further improve the manuscript, I do not see major concerns. As an independent additional reviewer I have the difficult task to not only judge the work of my peers, but also the work of a peer reviewer. Therefore I would like to ask the authors to carefully consider the minor comments I have, and address them appropriately, where possible. It is not my intention to impose more work, but I do think that my suggestions may further improve the manuscript for publication in Nature Communications, and some also agree with earlier comments raised.

We acknowledge the significant challenge presented by Reviewer #4's task and express our gratitude for their valuable comments and suggestions, which we enthusiastically welcome. We have diligently incorporated all of the excellent recommendations into the updated version of our manuscript, indicated by the use of blue highlighting.

Major concerns.

The authors have audaciously claimed that no method or reported probe exists for in vivo detection during aging or after senolytic interventions, conveniently ignoring the existence of well-documented references that have developed β -Gal-based probes for precisely that purpose (Nat Aging, 2023, 3(3):297-312; J Med Chem, 2021, 64(24):17969-17978; Anal Chem, 2020, 92(18):12613-12621).

We thank Reviewer #3 for his/her comments. However, note that we did not attempt to claim that “no method or reported probe exists for in vivo detection during aging or after senolytic interventions”. In the Introduction of our manuscript, we indicated: “However,

new strategies aimed at the detection of senescent cells and their monitoring during aging or after senolytic interventions are still required. Although the quantification of cell senescence in biopsies from different human tissues is an active area of research, non-invasive procedures to monitor cellular senescence in vivo have not yet been developed”. Note that the second sentence refers to humans, not to animal models. Our assertion does not suggest the absence of reported probes for in vivo β -Gal detection. Instead, it pertains to the absence of probes that can be retrieved from urine. In fact, the provided citations by the reviewer are good examples of probes that can be detected in vivo (in animal model), but not recovered in biological fluids.

- Nat Aging, 2023, 3(3):297-312: This paper describes a photoactivatable senolytic approach based on a system that is an enzyme substrate of senescence-associated β galactosidase (SA- β -gal).
- J Med Chem, 2021, 64(24):17969-17978: This paper described a probe that can be used in vivo non-invasively, similar to our WOS-Cy7Gal, but it is not excreted in urine.
- Anal Chem, 2020, 92(18):12613-12621: The same as the previous one.

We have provided a clearer explanation of the distinction in the updated Introduction (see highlighted text) and added the two last references suggested by the reviewer.

Your explanation here is clear. However, I was a bit confused by the highlighted added text. Especially the part ‘the signal detection relies on imaging methods’. Maybe also because it depends on what one considers non-invasive; some fins injection invasive, others do not even find imaging invasive, unless it is combined with an operation.

In my opinion, what you want to point out here is that there are certainly imaging methods available that make use of probes for senescence, but that you now developed a probe that can also be used without imaging modalities; by measuring it in collected urine. Measuring it in urine is an extra benefit. However if you only measure it in urine, you will not be able to pinpoint which tissue is most affected. As you can also use the probe for whole body imaging, and pinpointing the signal to specific tissues, as you show in the manuscript, I would suggest you highlight both these aspects. As you explain it well above, maybe you can use some of this text to rephrase the highlighted text?

Following the Reviewer’s suggestion, we have incorporated this aspect into the text (see text in the Introduction section highlighted in blue).

Furthermore, the authors fail to acknowledge that β -Gal is far from a specific biomarker for cellular senescence, and the presence of β -Gal activity during aging or senolytic therapy does not necessarily indicate the presence of senescent cells.

This has been a key point in the revisions. Reviewer #2 already indicated this in his/her second report: “describe what we are actually measuring is β -galactosidase activity- which could be associated with senescence”. Following that indication of Reviewer #2, we eliminated from the manuscript any reference to senescence-associated β -Gal activity and to senescent cell burden and indicated along the text that our probe measures β -Gal activity and that the probe readout can be correlated with changes during aging and senolytic interventions that also result in parallel changes in anxious behavior. We even changed the title of the manuscript to: “A renal clearable fluorogenic probe for the in vivo detection of β -galactosidase activity during aging and senolysis”. Therefore, note that in this last version we are not claiming in the manuscript that β Gal is a specific biomarker of cellular senescence.

This comment was addressed properly.

To add insult to injury, they attempt to tout the molecular scaffold of sulfo-Cyanine7 as a supposedly innovative near-infrared (NIR) fluorophore with improved water solubility, conveniently ignoring the fact that it is nothing more than a widely used and unremarkable component. As a result, this work contributes very little to the field of senescence detection or the design of molecular probes, rendering it virtually insignificant.

We never claimed in our manuscript that the molecular scaffold of sulfo-Cyanine7 is a supposedly innovative near-infrared (NIR) fluorophore, neither we claimed that sulfo-Cyanine7 has an improved water solubility. The presence of the sulfonic groups is to make the probe more diffusible and renally clearable, making the Cy7-Gal the first probe to detect β -Gal activity in biological fluids. As we mention in the text: "Cy7-Gal exhibited a more diffuse cellular pattern that was compatible with lower retention in lysosomes". Our probe is excreted and gives an organismal read-out of β -Gal activity in urine that correlates with age progression and is able to monitor the effects of senolytic treatment. In our perspective, this represents an advancement within the field of aging research. Note that this is an innovative approach and very few examples of urine-recovered probes have been reported, none of them for the in vivo monitoring of β -Gal activity. Moreover, note also that these renal clearable probes are closer to the potential translation into humans than other probes, which can be used also in a noninvasive manner, but require imaging.

Although not a central point of our manuscript, note that most available probes are mainly eliminated by the reticuloendothelial system, which can lead to their accumulation in the liver and spleen with the consequent potential organ toxicity and side effects. An alternative to this is the design of renal clearable probes which is an area of growing interest. Moreover, additionally in our case the probe, in an OFF state, is specifically transformed to an ON state by the action of β -Gal activity in cells/tissues in vivo, has a rapid elimination by the kidney into the urine where it can be easily measured.

In my opinion you have addressed this comment properly. I have to admit, upon first read of the manuscript, it was not entirely clear to me why first these sulfonic groups were omitted at first. If you add this sentence: 'The presence of the sulfonic groups is to make the probe more diffusible and renally clearable, making the Cy7-Gal the first probe to detect β -Gal activity in biological fluids.' either in the abstract or at the end of the introduction, this would make it more clear.

We thank the reviewer for this insightful comment, which has prompted us to recognize that the inclusion of two probes throughout the manuscript might confuse the readers. As per the reviewer's guidance, we have implemented adjustments to the text accordingly (**highlighted in blue; see pages 5-8**). Additionally, we have renamed the Cy7Gal probe to "**sulfonic-Cy7Gal**" to better align with the chemistry employed for achieving our objectives and to provide a clearer distinction from the cell-entrapped WOS-Cy7Gal.

Given these glaring shortcomings, I strongly advise against the acceptance of this manuscript in Nature Aging. At best, it may be suitable for publication in a more specialized journal, provided the authors diligently address the multitude of issues that still plague their work.

Major technical issues.

1. In this revised manuscript, the authors describe their probe to detect β -Gal activity during aging and senolysis, instead of senescence-associated β -Gal activity. Thus, the authors should improve the specificity of the probe for the detection of senescence.

Please see our comment above and previous comments by Reviewer #2.

Our probe is able to monitor β -Gal activity in vivo and its read-outs correlate with age progression and the effects of senolytic treatment. We agree with the reviewer that it would be interesting to develop a probe that detects senescence-associated β -Gal activity. However, this is not an easy task (and is out of the scope of our manuscript). The reason is that SA- β -Gal activity is not due to a specific β -Gal enzyme or a different lysosomal pH, but a higher activity resulting from increased lysosomal mass and increased expression of GLB1, the gene encoding the lysosomal β -Gal enzyme (Kurz et al., 2000, J. Cell Sci. 113:3613; Lee et al., 2006, Aging Cell 5:187). Therefore, a probe specifically generated to detect senescence is difficult to achieve based alone on β -Gal activity. In fact, most of the probes reported by now to detect senescence rely on the detection of the overexpressed lysosomal β -Gal enzyme in senescent cells (named as SA- β -Gal activity).

In my opinion this comment is properly addressed. I think this is a good point of discussion, and this should be added as such to the discussion section, as well as how you envision the probe to be used.

Following the Reviewer's suggestion, we have incorporated this aspect into the Conclusions section of our results, to accurately convey the limitations associated with using β -Gal activity as an indicator of cell senescence (see page 11).

2. In Fig 1b, a few senescent hUVECs have two Hoechst-labeled nuclei in one cell, which means that they still replicate. In this case, why does the fluorescence of WOS-Cy7Gal turn on in these cells?

We thank Reviewer #3 for this appreciation. It is true that a few hUVECs entering palbociclib-induced senescence are binucleated. It can also be seen in the X-Gal staining of hUVECs undergoing replicative senescence (Extended Data). To use another source of data unrelated to our work, the web page of the Cell Signalling company uses a picture of β -Gal histochemistry to advertise their Senescence β -Galactosidase Staining Kit that also stains binucleated cells (<https://www.cellsignal.com/products/cellular-assay-kits/senescence-beta-galactosidase-stainingkit/9860>). Binucleation or tetraploidy usually reflects endomitosis, i.e. cells enter M phase but exit prematurely, before completing cell division. Binucleated cells are found at very low frequency in healthy primary cell cultures, but their proportion can be increased depending on culture conditions. In any case, this cell state, as observed in our hUVEC cultures (hUVECs are primary cells), is not an indication that the cell is dividing but that it underwent endomitosis before entering palbociclib induced senescence. Interestingly, tetraploidy can also lead to senescence (Panopoulos et al., 2014, Mol Biol Cell 25:3105).

This is a valid explanation, and it would be very helpful if this would also be added in one or two sentences to the results discussion, as it does take away the question of why binucleated cells are present.

Following the Reviewer's suggestion, we have incorporated this aspect into the text, as part of the description of the cell senescent state induced by palbociclib (see page 4). We have decided to cite a classical paper with initial descriptions of senescent cells by the Hayflick

laboratory (Matsumura, T., Zerrudo, Z., Hayflick, L. Senescent human diploid cells in culture: survival, DNA synthesis and morphology. *J Gerontol.* 34(3):328-334 (1979)).

3. In Fig 1f, the immunostaining of p16 is not specific, which should be confined to nuclei and has no colocalization with nuclei. The authors should optimize the immunostaining of p16 and add the staining of nuclei.

Please see the type of expected staining as shown in the web page of the provider of the antibody used: <https://www.abcam.com/products/primary-antibodies/cdkn2ap16ink4antibody-epr20418-ab211542.html>. We can also provide citations in which a senescence-associated increase in the levels of p16 can be found in the cytoplasm and/or nucleus depending on the trigger (i.e., doxorubicin vs confluence). See, for example Spallarossa et al., 2010, PLoS One 5:e15583.

Specificity of p16 stainings has been an ongoing debate. Actually, only when you show no signal with the p16 antibody on p16ko tissue/cells would I be convinced that it is specific. However, in our lab we have used this antibody and it does seem to be specific as far as we can tell, at least for immunohistochemical tissue stainings. Most IF stainings I have seen show nuclear staining but some also show cytoplasmic staining. Therefore this is debatable. Would it be possible to add p21 staining as this should give a similar pattern as p16 and would support your data? Or do you have other stainings that would support the present p16 stainings? I do not want to impose more work, but it would be somehow good to be sure that we are looking at specific p16 staining.

A great deal of our experimental capacity to reach reasonable conclusions rely on elusive antibody specificity and the reviewer is completely right in that only ko tissue/cells can give someone that kind of assurance. We have used senescence-related markers other than the X-Gal histochemical reaction (Ki67, p16, γ H2AX, lamin B1, and p21) as per request of Reviewers #1 and 2, who considered that we needed to thoroughly characterize the senescent state in which we were claiming that our probe provided a clear read-out of the process. Some antibodies provide a clearer signal-to-noise ratio, but all of them have been used under the same conditions in non-senescent (control) and senescent cells and pictures have been taken with the same settings. In Figure 1, together with the stainings for p16, both by immunocytochemistry and cytometry (1e-h), we showed a quantitation of the changes in p21 staining in senescent vs. non-senescent hUVECs. We have now added images of the stainings for p21, lamin B1 and Ki67 used for the quantitations in **Supplementary Figure 1a**. It is true that this figure does not represent a colocalization of p21 immunofluorescence with WOS-Cy7Gal signal at the single cell level, but the quantifications show a correlated increase in both markers in the senescent condition. For the analyses at the single cell level, we chose p16 instead of p21 as the former is considered a more stable marker of the senescent state. In some cells, the increase in p21 is only transient.

4. In Fig 2a, the authors insist that the signals elsewhere other than in the bladder are background. In this case, the data processing should be optimized to minimize the background to increase the data accuracy. Moreover, representative pictures should be provided to account for the results.

We thank Reviewer #3 for his/her comment. Figure 2a illustrates the formulas of the compounds. Maybe the referee means to refer to Figure 2b. If this is so, this is a question already commented by the reviewer previously. In previous reports, we included representative images for the reviewer that illustrate the signal obtained in whole animals.

In this case, fluorescence can be detected at the site of the injection (intraperitoneal) and, in a palbociclib dose-dependent manner, in the tumors and the bladder (see Extended Data Figure 5a). Figure 2b shows representative pictures where fluorescence at the site of the injection is not shown and only emission in the lower belly of the animal (where the urinary tract is) is shown. Also note that all the data is coherent. Thus, as the hydrolyzed Cy7Gal probe is urinated it is expected that the in vivo emission in the animal (before mice urinate) comes from the bladder. Note that also the bladder has a large fluorescence ex vivo when the probe Cy7Gal, but not when the WOS-Cy7Gal probe is used (Figure 2f). This is coherent with the specific elimination of the hydrolyzed Cy7Gal probe in the urine. Nevertheless, and following the advice of the reviewer, we have revised Figure 2b and 3a, reducing the background of the IVIS® images to further limit the probe-associated signal and providing a representative image for each condition. Furthermore, we have delineated ROIs in the bladder area of BALB/cJyB mice bearing 4T1 tumors and treated with different doses of palbociclib, and also delimited this zone in BALB/cJyB mice from young and old age groups. We have quantified the fluorescence within these regions, and the corresponding data is now included in Extended Data Figure 5. We hope this can prevent any potential confusion.

In my opinion this comment is properly addressed.

5. In Fig 4g,h and Fig 5 e,f, the coefficient of determination R² is less than 0.5, which means there is a poor linear correlation between Cy7 fluorescence in urine and related factors.

Figure 4 illustrates our cytometry-based analysis and Figure 5e,f illustrate behavioral analysis in the elevated plus maze. We understand that the referee may be referring to Figure 5g,h and Figure 6e,f. Figure 5 g shows the significant linear correlation between Cy7 fluorescence in urine and the percentage of time spent in the center of the open field, in mice treated with D+Q or vehicle. Figure 5f shows the significant linear correlation between Cy7 fluorescence in urine and the percentage of time spent in the open arm of the EPM test in mice treated with D+Q or vehicle. Figure 6e shows the significant linear correlation found between the levels of Cy7 fluorescence in urine and anxious behavior in mice treated with D+Q or vehicle 15 days after the treatment, whereas Figure 6f, shows that there is not a significant correlation between Cy7Gal and anxious behavior when assessed in the long term after the senolytic intervention. Note that correlations with relatively low R² coefficients are normal when using behavioral tests. The positive R² coefficients found indicate a real relationship between the predictor (anxiety) and the response variable (Cy7 in urine) as indicated by the significance analysis.

6. The exponential powers of radiant efficiency between the IVIS imaging pictures and quantification data should not have such a big difference. For example, Fig. 2f (×10⁸) and 2g (×10¹¹), Fig. 3a (×10⁶) and 3b (×10⁹), Extended Data Figure 6 a (×10⁷) and b (×10⁹). The authors should double-check these data.

We have double-checked all the data.

I also do not understand this part; in Fig2g we see the quantification of 2f, where for the bladder we see a signal $\times 10^8$, for the tumor $\times 10^9$, and for the kidney $\times 10^7$. How then can the radiation efficiency be expressed in $\times 10^{11}$ for all these tissues in 2g? Could you clarify this? Maybe there is some conversion calculation that I have missed.

Following the IVIS®-Spectrum (PerkinElmer) user manual, we quantified fluorescence from IVIS® images by defining regions of interest (ROIs) to delineate fluorescent areas and measuring the average radiant efficiency for each ROI. To ensure accurate quantification, we deducted the corresponding background values related to animals (when analyzing *in vivo* fluorescence), Eppendorfs (when assessing urine samples), or any non-fluorescent regions (when examining *ex vivo* organs). The analyses were conducted using the Live Imaging software from Caliper Life Sciences. Regrettably, we have identified an issue during data transfer from the software to Excel files. This discrepancy occurred due to a language configuration mismatch, leading to the interchange of periods and commas (different use in English and Spanish). As a result, the recorded values appeared to be multiplied by a factor of 10³. We have now rectified this oversight. Thank you for your diligence in pointing out this issue.

7. In the revised manuscript, the author adopts an unusual form of figure representation, arranging bar charts and data dots separately. However, in Extended Data Figure 7, the “Cy7Gal” group’s data dots were not fit with the bar chart. This bar chart’s height was much lower than the data dots shown in the figure. We require the author to provide the specific values for dots and bars in the current Extended Data Figure 7. The author should clarify how this situation happened. Did the figure generated by the software automatically or have been adjusted manually?

This is a form of figure representation which we understand that it is quite common (at least it is common for us). We would like to stress that we have not adjusted any graph manually. We have shown all the data points and the bars represent the mean value in all the graphs. We have listed the date in the table below. If the Editor deems it appropriate, we can include these specific values for points and bars in Extended Data Figure 7.

	Cy7Gal		wo. Cy7Gal
	2-m	14-m	14-m
BLADDER	0,01	2,67	0,04
	0,28	1,81	0,03
	0,57	0,97	0,03
		0,48	
	mean	0,2867	1,4825
SD	0,2801	0,9635	0,0058
SEM	0,1617	0,4818	0,0033
BRAIN	0,26	1,18	0,25
	0,63	1,44	0,23
	0,54	0,51	0,07
		1,14	
	mean	0,4767	1,0675
SD	0,1930	0,3947	0,0987
SEM	0,1114	0,1974	0,0570
LUNG	0,09	1,11	0,06
	0	0,63	0
	0,42	0,07	0,09
		0,37	
	mean	0,1700	0,5450
SD	0,2211	0,4407	0,0458
SEM	0,1277	0,2204	0,0265

Normally one would show dotplots where the dots fall inside of the boxes. As an example:

In the figures of this manuscript all dots are outside of the boxes, this could be the source of confusion. My preference would be to have the dots inside the boxes. I leave it up to the editor to decide which is the right representation for the journal.

As for clarity, and following the reviewer's recommendation, we have now represented the dots inside the boxes. We had seen the representation of the dots in line outside the boxes in another work and opted to use it in our manuscript, but as most works show quantitative data with the dots scattered inside the boxes, our representation may confuse the readers, as indicated by the reviewer. Please, see **new figures**. We hope the modification enhances data visualization and comprehension of the readers. We have also changed the disposition of the bars in graphs shown in **Supplementary Figure 3a** (effects of the probe in cell viability) and in **Figure 2g** (fluorescence by imaging in tumors, kidneys and bladder) for clarity purposes. In addition, we have thoroughly reevaluated all graph data and statistics throughout the work.

8. In Extended Data Figure 3a, the control 4T1 group showed no positive β -gal staining results. However, several reported articles focused on 4T1 tumor detection strategies were based on β -gal detection. For example, [Evolving an Ultra-Sensitive Near-Infrared β -Galactosidase Fluorescent Probe for Breast Cancer Imaging and Surgical Resection Navigation. ACS sensors, 2022, 7(12): 3829-3837]. We believed that senescent 4T1 may have higher β -Gal levels than the control 4T1, but it doesn't mean there should be no X-gal staining positive region in the control group. Actually, in Extended Data Figure 3b, the control 4T1 cells + Cy7Gal probe showed a higher signal than the background, proving that there should be certain β -gal in normal 4T1. Too absolute results may mislead readers to think that there is no β -gal expression in 4T1.

This is an interesting question. There are multiple reported studies with the X-gal kit showing no positive staining results in normal cells and a clear positive staining in senescent cells when using staining. We have found very similar X-gal staining in normal and senescent 4T1 cells in other studies (see for instance Anal. Chem. 2021, 93, 3052). Note that no staining does not mean that there is not β -Gal, but that the β -Gal level is not enough to display a colorimetric signal taking into account the special conditions in which the X-gal kit is used (suboptimal pH for the β -Gal enzyme; Debacq-Chainiaux et al., 2009, Nature Protoc. 4, 1798). In addition, to use another source of data unrelated to our work, in the web page of the Senescence β -Galactosidase Staining Kit #9860 from Cell Signalling Technologies there is a picture showing no staining at pH 6 on normal WI38 cells at population doubling 29 but a clear blue X-gal staining of senescent WI38 cells at population doubling 36. In addition, note that it is very difficult to compare the degree of senescence, in the absence of an inducer, between our work and the cited one as the latter does not inform about number of cells per graft or time between transplantation and analysis. One has to keep in mind, however, that 4T1 are immortal cells of a tumor cell line and they do not enter senescence unless forced to do so (i.e. with palbociclib).

This comment has been properly addressed.

9. In Extended Data Figure 3e, the author tried to prove that the Cy7Gal probe had no cytotoxicity. However, at concentration 1.1 (log[Cy7Gal]) μ M, the viability of control 4T1 cells + Cy7Gal was only 60%, probably reaching the level of some chemotherapy drugs. The author should explain why this concentration was so unique because higher concentrations also showed no obvious cytotoxicity.

The data in Extended Data Figure 3 shows that the Cy7Gal probe exerts no detrimental effects on cell viability. As many as 10 increasing concentrations were tested. As mentioned by the reviewer, we found an apparent effect with one of the concentrations. We believe that the graph unmistakably illustrates the absence of cytotoxicity. Nonetheless, we have

generated a new graph. We have considered that the way we presented the concentrations of the Cy7Gal probe in this graph was a bit confusing, as we represented them as $\log[\text{Cy7Gal}]$. We have simplified the graph to more clearly indicate the concentrations used in the experiments and much higher: 25, 50, 100, and 200 μM .

I have not seen the original figure, therefore it is difficult to judge this adaptation. I do have an additional question; cytotoxicity is very important as you would like to use this probe in vivo, and maybe even in humans. Now there is a high clearance rate so long-term toxicity of the probe may not be an issue; have you looked at this, or any gathered any data on this? For the viability assay you use one that measures ATP. ATP content can be different between senescent cells and non-senescent cells. Have you measured cytotoxicity in any other manner, such as simply counting cells, or another type of viability assay with another outcome measure? I ask this as now in your graph the senescent cells seem more viable, which seems unlikely and may be due to the difference in ATP content. As you especially emphasize the non-invasiveness of the probe, any data you have on (non) cytotoxicity of the probe would be very helpful.

This is a very interesting comment. We employed the widely used cellTiter-Glo® Luminescent Cells assay kit (Promega), which indeed measures ATP content as a proxy for cell viability because is more sensitive than an MTT assay, which we also use regularly. Our data indicate that the sulfo-Cy7Gal probe does not have detectable cytotoxic effects in either replicating or senescent cells.

We have, however, taken into consideration the reviewer's comment, which has turned out to be very interesting. As with regards to potential changes in ATP content between senescent vs. non-senescent cells, the former are reportedly less efficient in producing ATP (increase in AMP/ATP ratio) due to reductions in mitochondrial membrane potential and OXPHOS efficiency (Zwerschke et al., 2003, *Biochem J.* 376,403–11; Hutter et al., 2004, *Biochem J.* 380, 919–28). Indeed, measurements in the viability test were lower in our senescent cultures (0.68 ± 0.03 % of the non-senescent culture values, $n = 3$). Because of this, we have now represented the viability, relative to the untreated condition, separately for replicative and senescent cells and thank the reviewer for pointing this out.

Regarding viability, we have also included our observations with the use of the probe in animals. The sulfonic-Cy7Gal was injected in 7 SAMP8 vehicle mice at 7-m before the senolytic intervention and again at 8, 9 and 11-m for assessing the treatment. On the other hand, 6 C57BL/6 mice were injected with the probe at 15-m, before senolysis, and then again at 17-m. All of them survived the entire senolytic longitudinal study, with no unforeseen health issues, until the moment of euthanasia at 12-m in SAMP8 and 18-m in C57BL/6 mice. Although this is partial information, and in the absence of systematic toxicity assays in mice, it conveys the idea that the probe is not overtly toxic for animals (see text in page 11).

10. In Extended Data Figure 3f caption, the author declared that “Note that while WOS-Cy7gal displayed a dotted pattern, Cy7Gal showed a more diffuse one”. Considering the two pictures were quite similar, we suggested the author use Image J to analyze the pictures. On a linear path, a “diffuse” pattern should be smoother than a “dotted” pattern.

We thank the reviewer for the suggestion. We have now applied image analysis to demonstrate that the pattern of cell staining is, indeed, different. We have added this

information as new Extended Data Figure 2f,g and a paragraph in Methods explaining how the image analysis was done.

This comment has been properly addressed.

11. In Extended Data Figure 4, mice carrying orthotopic 4T1 were treated with Palbociclib. 50 mg/kg or 100 mg/kg Palbociclib treatment led to a 50% tumor shrinkage on day 14. However, in Extended Data Figure 6a, mice treated with 50 mg/kg Palbociclib showed an even larger tumor size than the untreated group, whereas the 100 mg/kg Palbociclib treatment group had a smaller tumor size. The author should provide detailed information about when the IVIS experiment, the Ki67 staining and β -Gal staining were carried out in the Methods section. In addition, the author needs to confirm the tumor size in the IVIS image had no controversy when compared with the tumor growth curves.

We do not see any controversy between the tumor size in the IVIS image when compared with the tumor growth curves. Please note that as for tumor size, Extended Data Figure 4 shows measures with a caliper, whereas Extended Data Figure 6a shows dissected tumors and, therefore, the sizes may not be accurate due to resection effects.

This comment has been properly addressed.

REVIEWERS' COMMENTS

Reviewer #4 (Remarks to the Author):

All the comments have now been properly and vigorously addressed